# Energy and Macronutrient Dietary Intakes of Vegetarian and Semi-Vegetarian Serbian Adults: Data from the EFSA EU Menu Food Consumption Survey (2017–2022)

**DOI:** 10.3390/foods14081285

**Published:** 2025-04-08

**Authors:** Ivana Šarac, Jelena Milešević, Marija Knez, Marta Despotović, Marija Takić, Jasmina Debeljak-Martačić, Milica Zeković, Agneš Kadvan, Mirjana Gurinović

**Affiliations:** 1Center of Research Excellence in Nutrition and Metabolism, Group for Nutrition and Metabolism, Institute for Medical Research, National Institute of the Republic of Serbia, University of Belgrade, 11000 Belgrade, Serbia; jelena.milesevic@imi.bg.ac.rs (J.M.); marija.knez@imi.bg.ac.rs (M.K.); marta.despotovic@imi.bg.ac.rs (M.D.); marija.takic@imi.bg.ac.rs (M.T.); jasmina.martacic@imi.bg.ac.rs (J.D.-M.); milica.zekovic@imi.bg.ac.rs (M.Z.); mirjana.gurinovic@gmail.com (M.G.); 2Capacity Development in Nutrition CAPNUTRA, 11000 Belgrade, Serbia; k_agi@yahoo.com

**Keywords:** diet, vegan, vegetarian, semi-vegetarian, flexitarian, pescatarian, macronutrients, energy, protein

## Abstract

This study is the first to examine the diet and nutritional status of the adult vegetarian and semi-vegetarian population in Serbia, using data from the EFSA EU Menu Food Consumption Survey 2017–2022. The survey included 314 participants (63 vegans, 192 lacto-ovo vegetarians, 50 pescatarians, and 9 flexitarians), aged 18–74 years (166 women and 148 men, with no gender differences in dietary patterns) across all regions of Serbia. Collected data included anthropometrics (BMI) and intake of energy, macronutrients, and specific food groups (assessed through two 24 h dietary recalls). The study revealed multiple nutritional shortcomings across all three dietary patterns. The most significant was an insufficient protein intake (especially among vegans, but also among non-vegans), connected with an unsatisfactory protein quantity, quality, and availability in plant sources. There was also a high fat intake (particularly from omega-6 and trans-fats-rich sources), especially among non-vegans (but also among vegans), while the intake of omega-3 sources was low. Non-vegans consumed less carbohydrates, fiber, vegetables, and fruit, but more sweets, beverages, and alcohol. Our findings highlight the need for improved nutritional education of vegetarians/semi-vegetarians in Serbia and the development of national food system-based guidelines for this population.

## 1. Introduction

In recent decades, there has been growing interest in plant-based diets, with more individuals adopting plant-based eating patterns for diverse reasons, including health concerns, environmental considerations, animal welfare, ideology, cultural practices, religious beliefs, financial restrictions, socialization, or personal preferences [1,2,3,4,5]. Plant-based diets have been shown to reduce the incidence of various health conditions associated with increased morbidity and mortality [4,6,7,8,9]. Additionally, they contribute to lower greenhouse gas emissions from animal production [2,4,10,11]. Many religious practices also align with reduced or no meat consumption in general, such as those seen in Adventism, Buddhism, and Hinduism, or only during specific periods of limitation, such as Orthodox fasting [12].

Plant-based diets encompass a variety of dietary patterns, from strictly plant-based (vegan), to those that include dairy and/or eggs (such as lacto-, ovo-, or lacto-ovo vegetarian) and semi-vegetarian, which may include fish and seafood (pescatarian), poultry (pollotarian), or occasionally red meat (flexitarian) [1,9,13]. Definitions of semi-vegetarian patterns, particularly flexitarian, can vary substantially [6,9,10,13].

Many studies prove that vegetarian and vegan diets, compared to omnivorous diets, can offer numerous health benefits [6,7,8]. Vegetarians, vegans, and flexitarians/semi-vegetarians usually have lower body mass indexes (BMIs). When these diets are incorporated into hypo-caloric weight-reduction programs, they can lead to more significant weight loss [6,7,8]. Furthermore, vegetarians, vegans, and flexitarians often have lower serum levels of glucose, insulin, glycosylated hemoglobin A1c (HbA1c), total cholesterol, low-density lipoprotein (LDL) cholesterol, and high-density lipoprotein (HDL) cholesterol as well, with no notable difference in triglyceride levels or blood pressure (the latter decreased only among semi-vegetarians) [6,7,8]. Compared to omnivores, those following plant-based diets generally have a reduced risk of being overweight or having obesity, metabolic disorders (including diabetes and metabolic syndrome), cardiovascular diseases (such as myocardial infarction and stroke), chronic inflammatory diseases (like rheumatoid and inflammatory bowel diseases), and certain cancers (breast, prostate, and colorectal cancer) [6,7,8,14,15,16]. Additionally, these diets may result in better clinical outcomes during acute infectious diseases, as seen in studies on COVID-19 [17,18,19]. However, plant-based diets are also connected with lower bone density and an increased risk of fractures, as well as higher depressive and eating disorder symptoms [7,8,20,21,22]. While some trends suggest higher longevity in semi-vegetarian Adventists (regarding overall, cardiovascular, and cancer-related mortality), in vegetarians (regarding diabetes mellitus, cardiovascular, cerebrovascular, and chronic kidney diseases mortality), and in vegans (concerning ischemic heart disease mortality), conclusive evidence is lacking [7,8,23,24,25].

Nevertheless, not all plant-based diets are equally nutritious. Some may lack essential nutrients, such as proteins and specific amino acids (notably lysine, sulfur-containing amino acids, branched-chain amino acids—BCCA, and aromatic amino acids), omega-3 fatty acids, vitamins (especially B12 and D, but also A, B2, B3, and B6), trace elements (including iron, zinc, iodine, calcium, selenium, and copper), and antioxidants (creatine, carnitine, β-alanine, carnosine, taurine, ovotransferrin, and phosvitin) [8,9,22,26,27,28,29]. Moreover, they can contain anti-nutrients (like oxalates, phytates, tannins, catechins, lignins, lectins, saponins, goitrogens, glucosinolates, isothiocyanates, phytoestrogens, and inhibitors of trypsin, chymotrypsin, carboxypeptidases, elastases, α-amylase, and α-glucosidase), which may interfere with the absorption and metabolism of certain nutrients [8,9,26,27]. Still, these inhibitory effects are generally observed only at high intake levels and in individuals with preexisting nutritional deficiencies, and they can often be mitigated by food processing methods [26,27,30].

Additionally, many plant-based diets can be rich in ultra-processed foods, fried items, sweets, sweetened beverages, and other foods high in refined starches, added sugars (including high-fructose corn syrup), saturated and omega-6 fatty acids (like those found in coconut, palm, and sunflower oil), industrial hydrogenated trans and saturated fatty acids (in margarine and non-soy-based cheese substitutes), salt, additives, preservatives, artificial sweeteners, and alcohol [9,22,27,31,32,33]. All these components have been linked to increased morbidity and mortality associated with cardiovascular diseases, cardiometabolic syndrome, diabetes, fatty liver disease, and cancer [31]. Thus, choosing a diet that meets nutritional adequacy, health benefits, longevity, affordability, and sustainability remains a challenge [34,35].

Despite the prominent role of meat in traditional Serbian cuisine, with Serbians ranking among the highest meat consumers globally, plant-based dietary patterns are beginning to emerge even in Serbia [36,37,38,39].

Unfortunately, there is a lack of data on the characteristics of vegetarian or semi-vegetarian dietary patterns in the Serbian population.

The objective of this study, therefore, is to examine the dietary habits and nutritional status of the vegetarian and semi-vegetarian populations in Serbia, with a particular emphasis on their overall energy and macronutrient intakes, as well as energy and macronutrient intakes through specific food groups.

## 2. Materials and Methods

### 2.1. Study Design and Population

This nationally representative cross-sectional study, “The Serbian Vegetarian Food Consumption Survey”, was part of a larger national project aiming to examine dietary characteristics, nutrient intakes, and related health indicators among the overall adult population living in Serbia (“National Dietary Surveys in Compliance with the EU Menu methodology: The adults’ survey, including subjects from 10 to 74 years old”) [40]. All details regarding the overall survey aim, design, and methodology, including study population, sample size, sampling frame, ethical issues, study protocol, dietary assessment tools, anthropometric measurements, other data collected, software used, and staff training, are thoroughly described in reference [40], which also includes the ethical approval, the used written informed consent forms and questionnaires, and interviewer’s guidelines. Here, we will provide a brief overview focusing only on the parts relevant to the vegetarian dietary survey. In short, that overall survey involved 3018 individuals living in Serbia aged 10–74 years and included the subcategories of adolescents aged 10–17, adults aged 18–64, elderly aged 65–74 years, pregnant women aged 15–49, and individuals following the vegetarian/semi-vegetarian diet aged 18–74 [40]. The survey was designed following the European Food Safety Authority (EFSA) call for the acquisition of a harmonized pan-European Food Consumption database within the framework of the EU Menu process “What’s on the Menu in Europe?” (EU Menu), and followed the principles and standardized protocols established under the EU EFSA Menu methodology guidelines [41]. The study was conducted in accordance with the Helsinki Declaration and approved by the Ethics Committee of the Institute for Medical Research, University of Belgrade (Approval number: EO123/2017). Before inclusion into the study, the subjects were informed about the study objectives and procedures and signed the written informed consent for participation [40].

The present study intended to include a nationally representative cohort of vegetarians and semi-vegetarians (pescatarians and flexitarians) aged 18–74, across four geographical regions: Belgrade (capital city), Vojvodina, Šumadija and Western Serbia, and South-Eastern Serbia. As suggested in the EFSA call, since there is no official register of individuals following the vegetarian/semi-vegetarian diet in Serbia, the planned sample was considered a convenience sample [40], including a minimum of 260 valid subjects (130 per gender), with an additional 10% added to account for possible drop-outs [40]. Therefore, initially, the study planned to include 287 vegetarians/semi-vegetarians in a separate, “vegetarian” survey, 250 subjects aged 18–64 and 37 subjects aged 65–74 years [40]. Any vegetarians met during the main fieldwork were recruited until fulfilling the planned quota. However, for the purpose of this study, and to increase the study power, we also included the vegetarians/semi-vegetarians who were randomly recruited through other population surveys (i.e., the survey of adults, 18–64 years, and the survey of the elderly, 65–74 years) [40], leading to a total number of 314 valid subjects. Age categories (18–64, 65–74), gender, and geographical region were used as stratification criteria to warrant representativeness and diminish sampling bias [40]. Since there is no register providing information regarding the total number, gender, and geographical distribution of people following the vegetarian/semi-vegetarian dietary patterns in Serbia, stratification was performed based on the total Serbian population according to the Statistical Office of the Republic of Serbia Census data from 2011 [40].

The present study included only subjects who declared themselves as “vegetarians” (including vegans, lacto-ovo vegetarians, those not consuming any animal products, and those consuming only dairy products and/or eggs among animal products) or “semi-vegetarians” (including “pescatarians”, consuming fish at any frequency but consuming meat < 1 time/month, and “flexitarians”, also consuming meat < 1 time/week, 1–3 times/month) [3,42]. The subjects had to follow their declared dietary pattern for at least one year.

The subjects were recruited from private households across Serbia (individuals residing in institutional settings were excluded). Since vegetarian/semi-vegetarian practices are not common in Serbia, additional recruitment strategies were applied. For example, several vegetarian/vegan associations, vegetarian restaurants, yoga/healthy lifestyle community organizations, and Seventh Day Adventist Church associations were contacted and asked to share the information regarding the survey with their members [40].

### 2.2. Data Collection

Data were collected during 2017–2022, with interviews evenly distributed across all four seasons, to account for potential seasonal fluctuations in dietary patterns.

Trained interviewers conducted face-to-face interviews utilizing a structured, project-designed questionnaire pre-approved by the EFSA [40]. The general questionnaire included sociodemographic, lifestyle, and health-related information, including medical histories and medication usage. Data on the level of physical activity over the past week were collected through the International Physical Activity Questionnaire—Short Form (IPAQ-SF) [43]. Metabolic equivalents of the task (METs) (expressed in minutes per week) were calculated as described by Craig et al. [43], and according to the IPAQ-SF questionnaire results, physical activity levels were categorized as low, medium, and high [43].

Dietary intake data were obtained through two non-consecutive 24 h dietary recalls, at least one week apart, covering all days of the week [40]. Participants provided detailed information on their meals, consumed food items, dishes, and beverages, and their portion sizes [40], estimated from food packaging or by the validated Food Atlas for Portion Size Estimation for the Balkan Region [44]. Additionally, to check the habitual diet, the Food Propensity Questionnaire (FPQ) was applied to obtain the frequency of specific food groups/subgroups consumption over one year [40].

### 2.3. Data Processing and Dietary Assessment

Collected dietary data were analyzed using the advanced nutritional platform and software Diet Assess and Plan (DAP), evaluated by the EFSA for nutritional research [45,46], and integrated with the Serbian Food Composition Database (FCDB) for nutrient intake calculations [47,48]. In the FCDB, food items were categorized and coded according to the FoodEx2 classification system, to ensure harmonization with European datasets and cross-national comparability [41,47]. Composite dishes were disaggregated into individual ingredients to facilitate precise nutrient analysis, with yield and retention factors applied according to the European Food Information Resource (EuroFIR^TM^) guidelines [49,50].

Daily nutrient intakes were computed as averages derived from the two 24 h dietary recalls and expressed in both absolute units (e.g., kcal/day) and relative terms such as percentages of total energy intake (%TE). Since we do not have the Serbian national dietary guidelines, nutritional adequacy was assessed against the current EFSA (from 2017) and US (from 2020) age- and gender-specific dietary recommendations for the general population. The EFSA Dietary Reference Values (DRVs): Average Requirements (ARs) for energy [with physical activity level (PAL), set at 1.4, 1.6, and 1.8, to reflect inactive, moderately active, and active lifestyles], Average Requirements (ARs) and Population Reference Intakes (PRIs) for the intake of protein per kg of body mass, Adequate Intake (AIs) for dietary fiber, and Reference Intake Ranges (RIRs) for macronutrient contribution in total energy intake [51]. US Dietary Reference Intakes (DRIs): Estimated Energy Requirements (EERs), Estimated Average Requirements (EARs) and Recommended Dietary Allowances (RDAs) for protein intake per kg of body mass, Adequate Intakes (AIs) for dietary fibers, and Acceptable Macronutrient Distribution Ranges (AMDRs) for macronutrient contribution in total energy intake among the general population [52,53,54]. Additionally, we compared against the Kniskern and Johnston recommendations (from 2011) for protein intake per kg of body mass in the vegetarian population, which were adopted by the Italian Society of Human Nutrition (SINU) in 2017 [27,55].

### 2.4. Anthropometric Measurements

Body weight was recorded by the interviewers to the nearest kg using a calibrated portable digital scale, while height was recorded to the nearest cm using a portable stadiometer, employing the standardized procedures. The body mass index (BMI) was calculated as weight in kilograms divided by height in meters squared (kg/m^2^) and nutritional status was categorized based on the World Health Organization (WHO)/Food and Agriculture Organization (FAO) classification system [56]

### 2.5. Statistics

Data analysis was conducted using IBM SPSS Statistics for Windows software (v. 26.0., IBM Corp., Armonk, NY, USA). Distribution assessment of the numeric continuous data was conducted via the Kolmogorov–Smirnov test. Since all numeric data had a non-normal distribution, both in the total sample and subgroups based on gender and the type of vegetarian diet (vegan, lacto-ovo vegetarian, and semi-vegetarian), data were presented as *n* (%) for nominal data, and as the median and interquartile range (IQR) for non-normally distributed numeric data, with the non-parametric tests for hypotheses testing applied. The disparities in frequencies were tested through Pearson’s chi-square or Fisher’s exact test, with post hoc Z-tests adjusted by the Bonferroni correction for multiple tests. Differences between genders were tested using the Mann–Whitney U test. Differences between different dietary patterns were tested using the Kruskal–Wallis test with the post hoc Dunn’s test adjusted by the Bonferroni correction for multiple tests. Non-parametric one-way analysis of covariance (ANCOVA) was applied to test differences between different dietary patterns with adjustment for gender and age. The correlation between different continuous variables was tested by the non-parametric Spearman’s rank correlation coefficient. Statistical significance was denoted by *p* < 0.05 (in the case of multiple tests, *p*-values were adjusted with Bonferroni correction).

## 3. Results

### 3.1. Overall Sample and Gender Differences

#### 3.1.1. General, Socio-Demographic, and Physical Activity Data

The socio-demographic characteristics of the study sample are given in Table 1. A total of 314 vegetarians and semi-vegetarians were recruited. The study included slightly more women (~53%) than men (~47%), and the majority were lacto-ovo vegetarians (~61%), followed equally by vegans (~20%) and semi-vegetarians (19%), with no gender differences (vegans were slightly more represented among men, while lacto-ovo vegetarians among women, but that was not statistically significant). The median age was ~36 years, and most participants were less than 65 years old (it was particularly challenging to find those older than 65 years). One of the regions was a bit less represented (the South-Eastern Serbia region) but in accordance with the planned regional sample distribution [40]. Most of the examinees were from urban areas, of Serbian ethnicity, of Orthodox religion or atheists (but many Adventists were also recruited), single (not married), with a university education, without chronic illnesses and chronic usage of drugs (the most prevalent were diseases of cardiovascular and endocrine systems), never smokers, and moderately physically active (those with low or high physical activity made up about one-quarter each). All seasons were equally presented during the examination. There were no statistically significant differences between genders in the socio-demographic characteristics, except for marital status (more single men) and the presence of chronic diseases (more women, particularly with endocrine diseases), Table 1.

#### 3.1.2. Anthropometric Data

The median body mass was closely correlated with the median body height, leading to a median BMI within the range of desirable weight. Nevertheless, the BMI was higher in men (Table 2). The number of underweight subjects was more represented among women, while the number of overweight and obese subjects was more represented among men. All those gender differences in anthropometrics were highly statistically significant (*p* < 0.001 for all, Table 2).

#### 3.1.3. Total Energy and Macronutrient Intake Dietary Data

The median daily energy and macronutrient intakes are presented in Table 3. The median energy intake was higher in men (*p* < 0.001), and ~150 kcal lower compared to recommendations in both genders (~1850 vs. recommended 2000 kcal in women and ~2450 vs. recommended 2600 kcal in men), with an average of ~30 kcal/kg body mass in both genders. Protein intake per kg of body mass was below the EFSA recommendations for the general population (PRI, 0.83 g/kg body mass) in about half of the participants, but that was much more pronounced in women (~58% below recommendations) than in men (~45% below recommendations), *p* = 0.020 (comment: if we consider the US’ recommended RDA of 0.80 g/kg body mass, then ~55% of women and ~40% of men were below recommendations, *p* = 0.011). In about one-third of the population, the protein intake was even below the average requirements (AR, EAR) for 50% of the general population (0.66 g/kg body mass), which was again higher in women, but without statistical difference. On the other hand, if we consider recommendations for protein intake per kg of body mass for vegetarians (1 g/kg body mass), then the percentage with insufficient intake was even higher, ~68–69% in both genders, with no difference. Dietary fiber intake was below the minimal intake (25 g, according to the EFSA recommendations) in 30% of women and 10% of men (comment: if we consider the US recommendations, adapted per total caloric intake, then ~29% of women and ~30% of men were below recommendations, with no gender difference). The intake of all macronutrients (except alcohol) was higher in men (*p* < 0.001 for all). However, there was no gender difference in the macronutrient structure of the daily meal: carbohydrates dominated (~47%), followed by fat (~37%), while proteins provided ~11% of energy in both genders (Table 3).

When we analyzed the proportion of participants who satisfied the recommendations for energy intake and macronutrient composition of the diet (according to the EFSA and US guidelines), a high percentage in both gender groups had a total energy intake below recommendations (~30%, slightly more in women). Additionally, a huge percentage had insufficient protein and carbohydrate contribution (~40%), and excessive fat contribution (~60%). Women particularly had excessive fat contribution in their daily meals (Table 4).

#### 3.1.4. Specific Food Groups Contribution to Energy and Macronutrient Intakes 

Most energy originated from grains, followed by edible fats/oils, vegetables, fruits, and nuts/seeds. As expected, the caloric intake from these food groups was higher in men (Figure 1, Appendix A). However, there was no gender difference in the contribution of certain food groups to total daily energy intake, except for a negligible difference in the contribution of sweets (higher in women) (Table 5). In general, the contribution of animal sources to total energy intake was negligible in both genders, providing only 1.2% (0.0–9.3%) of energy in the total sample, 1.6% (0.0–10.7%) in women, and 0.6% (0.0–7.1%) in men (Table 5).

Grains were the main source of protein (both refined and whole grains), followed by vegetables (mostly legumes and fewer potatoes), nuts/seeds, and, to a lesser extent, fruits (particularly bananas), and their intake was (as expected) higher in men. In general, the contribution of animal proteins from milk products, eggs, fish, and meat to protein intake was negligible in both genders, providing only 0.2% (0.0–2.1%) of total energy, from 0.1% (0.0–2.0%) in men to 0.3% (0.0–2.4%) in women, with no gender differences, and almost all of that came from dairy products (Figure 2, Appendix A). In general, only 2.5% (0.0–14.8%) of protein came from animal sources in the total sample, with 4.2% (0.0–16.2%) in women and 1.7% (0.0–13.9%) in men.

The main sources of fat in the diet were edible fats and oils (with refined sunflower oil in first place, followed by olive oil and then margarines/butter, fat condiments, e.g., mayonnaise, non-soy-based substitutes for cheese, and other types of oils), nuts/seeds, bakery grain products, vegetable industrial products (e.g., tofu cheese, “humus”, “ajvar”), fruit (e.g., avocado), and cow’s milk cheese (Figure 3, Appendix A). Men had higher fat intake through edible fats/oils, nuts/seeds, and vegetables compared to women, and lower intake of fat related to plant substitutes for milk (e.g., soy, almond, oat, and rice-based beverages) (Figure 3, Appendix A). There was no difference between women and men in the fat contribution to total energy intake across different food groups, except for a negligible difference in the fat contribution from plant substitutes for milk, which was higher in women (Appendix A).

The main sources of carbohydrates in the diet were grains (mostly refined wheat, and less so whole grains—wheat, rye, refined and unrefined rice, oats, buckwheat, and breakfast cereals), then fruit (particularly bananas, and less so apples, watermelon, sweet cherries, oranges, and dates), vegetables (particularly tomatoes, potatoes, onions, and legumes—lentils and beans), sweets, beverages (cola and fruit juices), and alcoholic drinks (beer). Carbohydrate intake from grains, fruits, and vegetables was higher in men (Figure 4, Appendix A), but the percentile contribution to total energy was no different, except for dairy products and sweets, which were slightly higher for women (Appendix A).

### 3.2. Comparisons Between Different Vegetarian Dietary Patterns

#### 3.2.1. Socio-Demographic and Anthropometric Data

Since we identified three different eating patterns in our sample of vegetarians/semi-vegetarians, we examined the differences between vegans, lacto-ovo vegetarians, and semi-vegetarians in socio-demographics, anthropometrics, and dietary intakes.

Lacto-ovo vegetarians were slightly younger than vegans and semi-vegetarians, but other socio-demographic characteristics (from Table 1) were not different, except for geographical region, religion, and habits (Table 6). In Vojvodina and South-Eastern Serbia, there were more flexitarians and vegetarians, but fewer vegans, while in the Belgrade region and Šumadija and Western Serbia, there were many more vegans. Among vegans, there were many more Catholics compared to the other two groups. Compared to lacto-ovo vegetarians, among vegans, there were fewer current smokers and more former smokers, while flexitarians had a lower proportion of those who never smoked. However, many of these inter-pair comparisons lost significance due to the Bonferroni correction (Table 6). Vegans also tended to have higher physical activity levels, but again, the differences were not statistically significant. There was no difference in body mass, height, and BMI. The number of underweight subjects was slightly more represented among vegans, the number of overweight subjects among semi-vegetarians, and the number of obese subjects among lacto-ovo vegetarians, but those differences in anthropometrics were not statistically significant. (Table 6).

#### 3.2.2. Differences Between Dietary Patterns in Total Energy and Macronutrient Intake

There was no difference in total energy intake between the three dietary patterns, but the intake of protein was much lower in vegans compared to lacto-ovo vegetarians or semi-vegetarians (Table 7). The percentage of those who did not meet the EFSA/US ARs/EARs for 50% of the general population (0.66 g/kg body mass) was the highest among vegans compared to lacto-ovo vegetarians or semi-vegetarians (~46% vs. ~30%), while the percentage of those who did not meet the EFSA PRIs for 97.5% of the general population (0.83 g/kg body mass) did not differ between groups, even though it was again higher among vegans (~65% vs. ~46–50%), but at the marginal significance (*p* = 0.063) (when we consider the US recommendations of 0.80 g/kg body mass, it was, respectively, ~60% vs. ~45%, *p* = 0.045). Nevertheless, in all groups, the percentage of those who did not meet the recommendations for protein intake for vegetarians (1 g/kg body mass) was high, with no statistical differences, even though it was again the highest among vegans (~76%), then among semi-vegetarians (~73%), while among lacto-ovo vegetarians, it was the lowest (~65%). Vegans also had a much lower intake of fat, compared to lacto-ovo vegetarians or semi-vegetarians, and a higher intake of dietary fiber. A much higher percentage of those who did not meet the recommendations for fiber intake was among non-vegans (22–23, i.e., 33–36%, depending on the EFSA or US recommendations, respectively), compared to vegans (only 14, i.e., 13%). When energy from macronutrients was expressed as a percentage of total daily energy intake, the differences were even more significant: vegans, compared to lacto-ovo vegetarians and semi-vegetarians, had a much lower contribution of protein and fat, and a much higher contribution of carbohydrates and fibers in the energy composition of daily meals (Table 7).

In comparison with the EFSA and US guidelines for energy intake, no difference between the three dietary patterns was observed, even though vegans had the highest percentage of insufficient energy intake (~35%) (Table 8). In comparison with the US and EFSA guidelines for macronutrient composition of the diet, a much higher percentage of vegans had insufficient protein contribution in the diet (~62%), compared to lacto-ovo vegetarians and semi-vegetarians (~33% and ~36%, respectively). In contrast, a much higher percentage of lacto-ovo vegetarians and semi-vegetarians had excessive fat contribution (~63–71%) and insufficient carbohydrate contribution (~43–53%) compared to vegans. The differences in the macronutrient composition of daily meals were statistically significant (Table 8).

#### 3.2.3. Differences Between Dietary Patterns in Specific Food Groups Contribution to Energy and Macronutrient Intakes 

Among semi-vegetarians, nine subjects (15.3%) declared that they eat meat occasionally (1–3 times per month or less, i.e., “flexitarians”), and fifty-seven subjects declared that they eat fish (among them, “pescatarians” made up fifty, while seven occasionally consumed meat, i.e., “flexitarian”). However, only two subjects (3.4% of semi-vegetarians) reported eating meat during the interview days (~40 and ~50 g/day), while seventeen subjects (28.1% of semi-vegetarians) reported eating fish (maximum 200 g/day).

There was a significant difference between vegans and lacto-ovo vegetarians or semi-vegetarians in the total caloric intake from different food groups (Figure 5, Appendix A) and the contribution of different food groups to total caloric intake (Table 9). As expected, the main differences between vegans and the other two groups were in the intake of milk and milk products, eggs, meat, and fish (the latter two only applied to semi-vegetarians). However, there was also a much lower intake of edible fats/oils and sweets and a much higher intake of fruits and vegetables (the latter only applied to semi-vegetarians and when the data were expressed as a percentage of total energy). In contrast, lacto-ovo vegetarians differed from semi-vegetarians only in the intake of meat and fish, which was expected (even though the intake of meat and fish was negligible in semi-vegetarians, particularly the intake of meat). Lacto-ovo vegetarians also had a higher intake of grains and vegetables compared to semi-vegetarians, but that did not reach statistical significance (Figure 5, Appendix A).

Differences in protein intake from specific food groups followed the same pattern, with vegans having a much lower intake of proteins not only from animal sources, but also from sweets, fat condiments (e.g., mayonnaise), and plant-based substitutes for milk, and having a much higher intake of proteins from fruit (e.g., bananas) and vegetables (but with borderline significance). The differences between lacto-ovo vegetarians and semi-vegetarians existed only for meat and fish (Figure 6, Appendix A). Those differences also persisted when data were expressed as a percentage of total daily caloric intake and became even more obvious (Appendix A).

Differences in fat intake from specific food groups were similar, with vegans having a much lower intake of fat not only through animal products but also through sweets and edible fats and oils, while lacto-ovo vegetarians and semi-vegetarians differed only in fat intake from meat and fish (Figure 7, Appendix A). Differences from vegans persisted even when the data were expressed as the percentage of total daily energy (Appendix A).

Differences in carbohydrate intake from specific food groups were similar, with vegans having a much lower intake of carbohydrates from sweets and a higher intake from fruit compared to both lacto-ovo vegetarians and vegans, and a marginal difference in the intake of carbohydrates from grains compared to lacto-ovo vegetarians (higher in lacto-ovo vegetarians) and from vegetables compared to semi-vegetarians (higher in vegans), but this was not statistically significant (Figure 8, Appendix A). When the data were expressed as the percentage of total daily energy, results were similar, and differences became even more significant (Appendix A).

#### 3.2.4. Differences Between Dietary Patterns in Total Energy and Macronutrient Intake After Adjustments for Sex, Age, and Other Socio-Demographic Characteristics

In our sample of vegetarians/semi-vegetarians, we found a significant negative Spearman’s correlation between age and the intake of total energy, protein, fat, eggs, grains, sweets, alcohol/beverages, along with a positive correlation between fruit and BMI, with *p* < 0.01–0.001 for all cases. Since not only sex but also age could influence dietary intakes, and our dietary pattern groups were not completely equal regarding sex and age (lacto-ovo vegetarians were younger), we also performed comparisons of dietary patterns with adjustments for age and sex by applying the non-parametric ANCOVA test. All significant differences between vegans and lacto-ovo vegetarians/semi-vegetarians remained significant even after adjustment for age and sex.

In our study sample, we did not find any significant associations between dietary intakes or dietary patterns and socio-demographic characteristics such as level of education, settlement type, religion, marital status, household member number, and existence of chronic diseases when age and gender were taken into account (by performing partial Spearman’s rho correlations adjusted for age and gender). We only found positive associations between smoking habits and total energy, fat, carbohydrate, alcohol, dairy product, egg, added fat, oil, grain, and non-alcoholic and alcoholic beverage intakes, and negative associations with fiber and fruit intakes, while positive associations between physical activity levels and fruit, nut, seed, and occasional meat intakes, and negative associations with added fats/oils and grains intakes, were also found. However, differences between vegans and lacto-ovo vegetarians/semi-vegetarians remained significant even after additional adjustments not only for age and gender but also for smoking and physical activity levels in the non-parametric ANCOVA tests.

## 4. Discussion

The present study provides the first comprehensive analysis of energy, macronutrients, and specific food group dietary intakes among vegetarians and semi-vegetarians (including pescatarians and flexitarians) residing in Serbia, differentiated by gender and dietary patterns. The survey involved a nationally representative sample from all regions of Serbia, conducted across all seasons, and utilized the validated EU Menu methodology, which aligns with the approaches used in other European countries. The study revealed numerous nutritional inadequacies regarding energy and macronutrient intakes within this population.

Among 314 recruited vegetarians and semi-vegetarians, slightly more were female (~53%) and most were lacto-ovo vegetarians (~61%), while the rest belonged similarly to vegan and semi-vegetarian dietary patterns. The semi-vegetarian dietary pattern mostly resembled the pescatarian diet (in ~85% of semi-vegetarians), while only 15% of them also reported eating meat occasionally (but no more than 1–3 times per month). One region was slightly less represented (the South-Eastern Serbia region), in accordance with the lower number of citizens, according to data from the Statistical Office of the Republic of Serbia from 2011 [40]; but, in that region, there is a typically lower occurrence of vegetarian dietary patterns.

Unfortunately, there is no official data on the number of individuals who follow the vegetarian/semi-vegetarian dietary patterns in Serbia, but some data indicate that, in general, there is a much lower percentage of vegetarians/semi-vegetarians in Serbia than in other parts of the world; according to the Ipsos survey conducted in 2018, including 20,313 adults across 28 countries over the world (using the Ipsos Online Panel System on the Ipsos Global Advisor platform) [38], Serbia was one of the top meat-eater countries in the world, together with Hungary, having ~91% of the population reported to be omnivorous (Russia was in third place, with 88% reported to be omnivorous). According to the same research [38], globally, 73% of the world population follows the omnivorous dietary pattern, while 14% is flexitarian, 5% vegetarian, 3% vegan, and 3% pescatarian. An omnivorous type of diet was more often reported over the globe among males (which is in accordance with eating meat being perceived as more “masculine” [3]), those older than 35 years, and high-income countries and households [38]. In Serbia, however, there is often an antagonism toward vegetarianism (particularly veganism), and it is often perceived as “anti-traditional”, “western”, an attack on traditional values, “non-masculine”, or unhealthy [36,37], although during the traditional religious fast, plant-based or pescatarian dietary patterns are employed [37]. Since the vegetarian dietary pattern is quite “new” in Serbia (meat traditionally has an essential role in Serbian cuisine) [37], it is expected that younger generations are more likely to adopt vegetarian and semi-vegetarian patterns, which is in accordance with the findings from other studies [57].

The nutritional adequacy of vegetarian and semi-vegetarian diets in countries across the globe has been assessed in multiple studies [9,22,31,32,33]. Although in Western, high-income countries vegetarian diets (including vegan) have proven to be safe and nutritionally well-balanced, in some lower-income countries, they have been shown to increase the risk of certain nutrient inadequacies, possibly leading to hidden nutritional deficits [5,22,27,31,32,58,59,60]. The most prevalent are micronutrient deficits (particularly of vitamins B12 and D, iron, zinc, iodine, calcium, and selenium), but there are also concerns about the adequacy of total energy and some macronutrient (particularly protein and omega-3 fatty acids) intake [22,27,32,33].

The present study revealed a slightly lower energy intake compared to the recommended amount (particularly in vegans and women), and a significantly lower protein intake compared to the recommended amount (again, particularly in vegans and women). Although the median intake of energy was slightly lower (~150–200 kcal) than generally recommended for age, gender, height, desirable body weight, and physical activity level by the FAO/WHO, EFSA, and US Institute of Medicine (IOM) [51,52,61], the body mass was adequate in the majority of the subjects, and the median BMI values were within the optimal range. However, the number of underweight subjects was the highest among vegans and women (9.5% and 8.4%, respectively), which was much higher than in the general adult population in Serbia (the results from the same survey in the general adult population, *in press* [39,40]). In contrast, the number of overweight/obese subjects was the highest among semi-vegetarians and men (27.1% and 31.1%, respectively), but still lower compared to the general Serbian population [39,40].

In our study, the physical activity levels were the highest among vegans, which can explain the highest prevalence of underweight subjects and the lowest prevalence of overweight/obese subjects in this subgroup, despite similar median energy intakes. In contrast, compared to men, women had a lower energy intake and the same level of physical activity, which resulted in generally lower BMIs.

Other studies [58,59,62,63,64] (but not all [22]) also pointed out the lower energy intake in vegetarians and semi-vegetarians compared to omnivores, resulting in lower BMIs and a lower prevalence of overweight and obese individuals [7,8,58,64,65]. Nevertheless, in our study, the energy intakes were similar or somehow lower compared to other Western (European and US) countries (where the majority of the included vegetarians/semi-vegetarians were females) [62]. In contrast, data from some other surveys (e.g., National Health and Nutrition Examination Surveys—NHANES) indicated considerably lower energy intakes among US vegetarians compared to Serbian vegetarians (e.g., 1750 kcal vs. 2050 kcal), despite much higher BMI in the US cohort (28 vs. 22.6 kg/m^2^), which can reflect discrepancies in the study methodologies and possible underreporting in the US cohort [63]. Also compared with the UK Biobank study, men in our study had a much higher energy intake vs. only ~2000 kcal reported in that study, despite the much higher BMIs in the UK study (~25 kg/m^2^), indicating possible dietary underreporting [58].

In our population, the average energy intake per kg of body mass was ~30 kcal/kg, which is lower than the recommended value for age, gender, and the level of physical activity (~36 kcal/kg) [52,61]. However, due to the possible underreporting of dietary intakes [66,67,68], day-to-day variability in food and nutrient intakes [69], and possible imperfections in our dietary tools, we cannot claim that the total energy intake was insufficient in our population (considering BMI). Additionally, since carbohydrates and fat are the main sources of energy in the body [26,70], and their intakes were sufficient in our study (particularly the intake of carbohydrates in vegans and the intake of fats in lacto-ovo vegetarians and semi-vegetarians, which was even above recommendations), it was unlikely to develop an energy deficit. Dietary fats had the highest metabolic efficiency [71,72]. An adequate intake of carbohydrates can spare proteins from their catabolism [73,74]. Unfortunately, due to methodological limitations, we did not measure body composition to see if there was a lower percentage of lean body mass and muscle mass among our vegans, lacto-ovo vegetarians, and semi-vegetarians, which would be an important anthropometric indicator of nutritional adequacy.

In our study, the macronutrient structure of the daily meal was dominated by carbohydrates (~47%), followed by fat (~37%), while proteins provided ~11% of energy in both genders, which is not completely aligned with recommendations (particularly regarding fat intakes) [51,52,71]. However, the vegan diet was particularly inadequate regarding the contribution of protein to total energy (~62% of vegans had protein contribution below 10% of total energy, on average 9.1%). In contrast, lacto-ovo vegetarians and semi-vegetarians were characterized by the increased contribution of fat and decreased contribution of carbohydrates in the diet compared to vegans, better resembling the overall dietary omnivore pattern in Serbia [39]. Our data regarding the contribution of protein in the energy structure are lower compared to the data from other European countries or the US, where proteins contribute to 13.7–16.5% of total energy among flexitarians, 14.5–15.5% among pescatarians, 10.0–16.4% among lacto-ovo vegetarians, and 11.1–14.1% among vegans [33,62,75,76].

The main sources of energy in our study were grains and their products (~26%), followed by edible fats/oils, vegetables, fruits, and nuts/seeds, while the main sources of proteins were similarly grains and vegetables (mostly legumes), and less nuts. In the total study sample (including vegans), animal sources (mainly from dairy products) contributed only ~1.2% of energy, while animal source proteins only contributed ~2.5% of proteins and ~0.2% of total energy. Even when only semi-vegetarians and lacto-ovo vegetarians were considered, animal sources contributed only ~3.0–3.4% of energy, while animal source proteins contributed only ~9.0–9.5% of proteins and ~0.7–0.8% of total energy, respectively. These data are very different from Western countries (e.g., the UK), where for lacto-ovo vegetarians and semi-vegetarians, animal sources provided about 10–15% of energy [58], while protein from animal sources provided 2.6–5.4% of total energy [2].

Plant-based diets (particularly those strictly based on protein, like veganism) are accompanied by the risk of protein and essential amino acids deficits due to generally lower protein intake in vegetarians [62] (since animal sources are the richest in terms of protein abundance) [26,77] and lower quality of proteins caused by an insufficiency of certain essential amino acids. For example, grains mostly lack lysine, but some also some leucine, threonine, and tryptophan; legumes mostly lack sulfur-containing amino acids methionine and cysteine, but some also BCCA—leucine, valine, lysine, tryptophan, and threonine; nuts and seeds lack methionine, lysine, valine, leucine, isoleucine, cysteine, and threonine; potatoes lack histidine; pseudocereals—quinoa, amaranth, and buckwheat—lack lysine (though less so than cereals), methionine, leucine, isoleucine, valine, and threonine) [77,78]. There is also a lower digestibility of plant proteins (0.50–0.99%) [27,77] because of different protein structure (protein cross-linking and protein chain rigidity) [26], the existence of anti-nutrients (e.g., inhibitors of trypsin, chymotrypsin, carboxypeptidases, elastases, phytates, glucosinolates, isothiocyanates, tannins, lectins, and saponins), and high fiber content, which all contribute to decreased protein digestion and absorption [26,27]. Additionally, plant proteins have a lower anabolic capacity, particularly during the postprandial period [26,62,77,78,79]. For that reason, since current recommendations for protein intake are based only on “good quality proteins” [80,81] and to cover lower quality and digestibility (assessed by the Protein Digestibility Corrected Amino Acid Score (PDCAAS) or Digestible Indispensable Amino Acid Scores (DIAASs) [77]), some authors have recommended higher protein intakes for plant-based diets (particularly in vegans and when plant proteins are above 70%), at 0.96–1.00 g per kg of body mass, which has been adopted by some nutritional societies (e.g., Italian) but not by all (e.g., American, Canadian, French, British, etc.) [27,32,52,82,83,84,85,86,87]. In fact, most of the position statements of national nutrition societies claim that with an adequate combination of complementary plant protein sources, it is possible to achieve an adequate intake of all essential and non-essential amino acids within the recommended total protein intakes for the omnivorous population, keeping in mind that in the vegetarian population, animal sources could still cover ~30–50% of proteins [54,62,77,79,80,88]. On the other hand, in our study, the contribution of animal sources even in non-vegans was only 9.0–9.5%, significantly below 30%, indicating a dietary pattern with compromised protein quality [88]. Despite this, some novel findings indicate that a vegan dietary pattern, particularly if prolonged, requires higher protein intake to achieve the amino acid balance, and that achieving sufficient protein and essential amino acid intake among vegans is challenging (particularly for lysine, and less so for BCCA—leucine, isoleucine, and valine) [26,82,88,89,90]. Furthermore, even for the general population, there are some concerns about the adequacy of the existing recommendations, with some authors recommending a higher intake of 0.87–0.93 g of protein per kg of body mass [91]. Additionally, it is challenging to effectively combine complementary plant protein sources in each meal, or even throughout one day, thus the adequate meal distribution of protein deserves further research [26,27,77]. According to one study, it is acceptable to have a ~3 h gap between meals with separated complementary proteins [77]. It is also recommended to split proteins equally among daily meals (e.g., to consume ~20 g of protein per meal, across a minimum of four meals), to achieve maximal digestion, absorption, and anabolic efficiency (e.g., after endurance training) [92], but there is also evidence that people who do not exercise benefit equally from having all their daily proteins in a single meal [92,93,94]. In general, it advisable to split plant proteins evenly (e.g., 20–30 g) throughout the day (each ~3–5 h) to increase anabolic efficiency, but more research is needed in that direction [26].

In our study, surprisingly, about one-third of the subjects did not meet the critical intake of 0.66 g of protein per kg of body mass for the general population (~28% of men and ~36% of women; among vegans even 46%), while about half of the subjects (and 65% of vegans) did not meet the general requirements for the adult population (0.80–0.83 g of protein per kg of body mass, depending on the guidelines). The recommendations for vegetarians, with a plant protein contribution of above 70%, and vegans (1 g per kg of body mass) were not met by ~70% of the subjects (again, the most among vegans, about three-quarters). The observed protein intakes per kg of body mass were much lower compared to other Western countries (UK, France, Belgium, Denmark, USA) [57,59,62,65,75,95,96,97]. For example, in the EPIC-Oxford study, the authors found a protein intake below 0.66 g per kg of body mass in only 10% lacto-ovo vegetarian men and 6% lacto-ovo vegetarian women [59], while in our study it was discovered in ~28% of lacto-ovo vegetarians (among whom 56% were women and 44% were men). In the EPIC-Oxford study, only 16.5% of vegan men and 8.1% of vegan women had a protein intake below 0.66 g per kg of body mass, while in our study, among vegans (among whom 57% were men and 43% were women), a total of 46% had intakes that low [59]. One study on German vegans found that 31.3% of males and 41.4% of females had protein intakes below 0.8 g per kg of body mass (with average intakes of 0.92–0.95 and 0.75–0.92 g per kg of body mass in males and females, respectively), while in our study, 62% of vegans had protein intake below 0.8 g per kg of body mass, and the median intakes were lower (0.69 g per kg of body mass) [97]. In the Nutrinet-Santé Study, 15% of lacto-ovo vegetarians and 27% of vegans had a protein contribution below 10% of total energy, while in our study, 33% of lacto-ovo vegetarians and even 62% of vegans had insufficient protein contributions [57]. This indicates that our vegetarian/semi-vegetarian population significantly differs from Western high-income countries by the much lower total intakes of proteins, proteins from animal sources (in non-vegans), and proteins from plant sources (in vegans).

Despite our awareness that protein intake is often underreported (compared with the nitrogen balance studies) [66,67,68], with possible day-to-day variability in protein intakes [69], this study highlights a concern that protein and amino acid intakes can be inadequate in our vegetarian/semi-vegetarian population, particularly in vegans. Even in lacto-ovo vegetarians and semi-vegetarians, a high percentage (~30%) did not meet the critical intake, ~50% did not meet the optimal intake for the general population, and ~70% did not meet recommendations for vegetarians. This becomes even more alarming when we consider that only 0.7–0.8% of total energy and only ~9.0–9.5% of total proteins were from dairy products and eggs (mainly dairy products) in semi-vegetarians and lacto-ovo vegetarians, while the intake of protein and protein-related energy from meat and fish in semi-vegetarians was also negligible (tended to be zero in most of them). For that reason, we could consider the diet of semi-vegetarians and lacto-ovo vegetarians to be firmly plant-based, and that the majority of protein even among semi-vegetarians and lacto-ovo vegetarians came from plant sources. This indicates that the guidelines on the amount, combination, and distribution of plant proteins from different plant sources are also very important for our lacto-ovo vegetarians and flexitarians. Unfortunately, due to limited resources, this study did not include an assessment of biological indicators of the body status of proteins and essential amino acids, which would confirm our findings. Additionally, our FCDB does not cover the amino acid composition of the food.

The intake of not only proteins but also other macronutrients deserves attention in our study. Particularly among non-vegans, we found an increased contribution of fat in the diet. Fat intake originated mostly from vegetable oils, particularly sunflower oil, which is rich in omega-6 fatty acids [98]. The sources of plant omega-3 acids, i.e., alpha-linolenic acid (ALA) (flaxseed oil, flaxseeds, chia seeds, and walnuts) [99], were much less represented in the diet. As noted, the fat contribution of fish (which is the best source of physiologically important very-long-chain omega-3 acids) and eggs (which can be enriched with very-long-chain omega-3) was negligible and tended to be zero (even among lacto-ovo vegetarians and semi-vegetarians). In addition, studies have shown that the conversion of ALA to more physiologically important omega 3 eicosapentaenoic acid (EPA), docosapentaenoic acid (DPA), and docosahexaenoic acid (DHA) in humans is low (e.g., 0.2–2–8% to EPA and 0.01–1–5% to DHA, depending on the examined lipid body compartment and tracer technique used), and dependent on many factors (including sex, age, nutritional status, diet, physical activity level, smoking, etc.) [100,101,102,103,104,105]. In general, higher dietary intakes of ALA can increase EPA levels, but the effect on DHA levels is negligible in most compartments, and decreases have even been shown in some studies [105]. The increased ratio of omega-6 to omega-3 fatty acids in the diet and body compartments has not only been proven to increase the risk of many inflammatory, cardiovascular, metabolic, degenerative, and malignant diseases [102,104,106,107] but also to be negatively associated with the conversion of ALA to EPA/DHA, with the ratio > 4:1 being associated with worse conversion [103,108,109]. Therefore, diets rich in omega-6 fat sources (e.g., sunflower oil, as shown in this study) could further diminish the conversion of ALA to EPA [103,110]. In agreement with this and according to preliminary data from another recent study on Serbian vegetarians performed by our team (which included 80 omnivores and 80 vegans/lacto-ovo vegetarians from the Belgrade region), compared to omnivores, Serbian lacto-ovo vegetarians and vegans had higher proportions of oleic acid and omega-6 linoleic acid (LA) in their erythrocytes, but lower proportions of omega-6 gamma-linoleic acid (GLA), omega-3 EPA, DPA, and DHA, as well as total omega-3 acids, omega-3 index, and ratios of omega-3 to omega-6, ALA to LA, EPA to arachidonic acid (ARA), and DHA to ARA [111]. In that study, the ratio of omega-6 to omega-3 fatty acids in Serbian omnivores was 6.0 (significantly above the recommended <4.0 [108]), while in lacto-ovo vegetarians, it was 8.4, and in vegans, it was 10.5, almost doubled. The omega-3 index in omnivores was 4.0 (much below recommended  >6 [112]), while in vegans/vegetarians, it was only 2.1 [113]. In addition, they had lower estimated activity of delta 6 desaturase (D6D), the rate-limiting step in the ALA to EPA conversion [105,114], also involved in the conversion from EPA to DHA [114]. The findings from that study support the dietary findings from the present study, indicating that Serbian vegetarians are at an increased risk for omega-3 deficit, and are in accordance with the studies of vegetarians from other countries [65,83,103,110,115,116,117,118,119,120]. (Note: in our study, only 13 subjects reported consumption of fish oil omega-3 supplements 1–7 times per week, but the amount of EPA+DHA was up to 300 mg per day, which is probably not enough according to more recent research [121].)

Apart from sunflower oil, the other main sources of fat were olive oil (two times less than sunflower oil), hydrogenated plant oils (margarines), and other industrial products full of hydrogenated fats (including non-soy-based cheese substitutes), which are important sources of harmful industrial trans fatty acids, associated with increased risk for metabolic, cardiovascular, inflammatory, and malignant diseases [122,123]. This means that even if the diet of vegetarians and semi-vegetarians in Serbia is based on plant fats, it still includes the intake of unhealthy, hydrogenated saturated and trans fatty acids [123]. Cow’s milk fat was also an important contributor to fat intake among non-vegans, particularly in semi-vegetarians. Cow’s milk fat contains saturated fatty acids (including short-, medium-, long-, branched- and odd-chain saturated fatty acids), trans fatty acids (trans vaccenic acid, trans palmitoleic acid), and conjugated cis-trans linoleic acid—CLA, for which the health effects are still not well determined [122,124,125]. Additionally, it contains only small amounts of omega-3, mainly ALA, but a good ratio of omega-3 to omega-6 [126]. Even though there is still controversy about the health effects of milk fat, most of the US and European medical associations for the prevention of cardiovascular diseases continue to promote the consumption of low-fat dairy products in their current recommendations [124,125]. Unfortunately, we did not include the data on fatty acid profiles in diets in our analysis, due to the temporary incompleteness of our FCDB for such data (particularly for industrial products) and our software, which needs improvement to provide more detailed data for specific food sources per examinee. Additionally, due to methodological limitations, in this survey, we did not collect biological specimens to analyze the status of the specific fatty acid profiles in the body.

The intake of carbohydrates was adequate among vegans, but less adequate among lacto-ovo and semi-vegetarians, not reaching the recommended contribution in the diet (45%TE) in 43% of lacto-ovo vegetarians and 55% of semi-vegetarians. The intake of dietary fiber (although higher than in other Serbian adult populations [39]) was inadequate in about 30% of the total sample, but only in 13% of vegans and 33–36% of non-vegans. Refined grain products were the largest contributors to carbohydrate intake (particularly white bread and refined wheat flour products), but some of the participants also included non-refined grains and pseudocereals. Vegetables (tomato, potato, onions, and, to a lesser extent, legumes), and fruit (particularly bananas, and less so apples) were also important contributors to carbohydrate intake, particularly in vegans. However, sweets and sugar-added beverages (e.g., cola) also contributed, primarily among non-vegans. These results highlight distinct carbohydrate intake patterns between vegans and non-vegans: non-vegans had a higher intake of refined sugars, which are associated with health risks, while vegans consumed significantly more complex carbohydrates and fiber, which are linked to health benefits [127]. Unfortunately, at present, our FCDB does not provide information on added sugar content in all food items, and therefore such analyses were not obtainable.

Interestingly, another important contributor to carbohydrate intake was beer (belonging to the top 25 contributors), particularly among non-vegans. The intake of alcohol (particularly from beer) was the highest among non-vegans and men (as expected) but, in general, was very low in the whole sample.

This study has several strengths and limitations. Strengths include a well-designed study protocol, representatives of our stratified sample across geographical regions, genders, age groups, and seasons, and the use of validated tools in data collection and analysis, approved by the EFSA, which allows comparison with other European countries. The study includes overall information, but also both gender and dietary pattern comparisons. However, there are some limitations. Firstly, we do not have data on the percentage of vegans, lacto-ovo vegetarians, pescatarians, and flexitarians in the overall adult population of Serbia, and we are not sure if our sample size adequately resembles each dietary pattern. Because of the lack of such data, the required sample size was defined in accordance with the convenience sampling procedure, following the EFSA call and guidance on the EU Menu methodology design [40]. A further limitation is the usage of two 24 h dietary recalls, which may not represent long-term dietary habits and can introduce potential recall bias and misreporting because they rely on memory, cognitive performances, literacy, education, subjects’ motivation, the potential aspiration to give desirable answers, and the ability to estimate the portion sizes accurately. However, we applied several measures to reduce all potential reasons for improper data collection. First, we additionally applied the FPQ to account for habitual diets over one year [40]. This confirmed the representativeness of the 24 h recalls, and our interviewers were well-trained and instructed not to collect any non-representative data for one’s habitual diet. Second, in order to capture the most realistic data and avoid misreporting during the 24 h recalls, the interviewers were instructed to use the multiple-pass questioning method, with the application of combined structured and unstructured interview steps and probing questions (e.g., on food leftovers), as already described in detail in reference [40]. Third, we used a validated food picture book—The Food Atlas for Portion Size Estimation for the Balkan Region [44], as well as information from food packages, to help participants and interviewers determine portion sizes. Some other studies in European general populations have indicated that there can be an underestimation of protein intakes of about 10% for two dietary recalls (compared to nitrogen excretion studies), particularly with higher intakes and when the subject’s BMI is elevated (similarly for energy intakes and other nutrients; however, high protein intake and high BMI were not typical for this population), and that there can be day-to-day variability in macronutrient intakes [66,67,68,69]. Therefore, further studies are needed to assess the appropriateness of protein dietary intakes in this population, implementing other complementary techniques. Another limitation is the lack of information on specific fatty acids, amino acids, and added sugar consumption in our analysis, due to the temporal paucity of such information for all food items in our FCDB (which encompasses ~2000 food/dish items), particularly for industrial products, so further improvement of our FCDB will also enable such analyses. An additional limitation is that due to the overall EFSA project design in accordance with the EU Menu methodology [40], only the BMI was used as the biological indicator of nutritional status, due to limited available resources for such a large sample. It would be ideal to also have data on body composition (e.g., fat mass, fat-free mass, muscular mass) and other anthropometric and biological (e.g., biochemical) indicators of the adequacy of nutritional statuses for macronutrients (e.g., status of protein, amino acids, omega-3 fatty acids, etc. in the body), to compare with our dietary data. Therefore, further research and a more detailed examination of this population are needed to support our dietary findings. Currently, we are conducting a study involving 80 vegans/lacto-ovo vegetarians and 80 omnivorous participants from the Belgrade region, where blood samples are also to be collected for various health and nutritional assessments. Finally, due to the absence of Serbian national dietary guidelines for energy, macronutrients, and specific food group intakes, we compared our results on energy and macronutrients with the current EFSA and US recommendations (which slightly differed for some items), finding these the most relevant to Serbian dietary patterns. We chose these recommendations over others, such as those from the WHO, because they better reflect the general dietary habits in Serbia, which are more similar to those of Western countries, with higher intakes of fats and proteins and lower intakes of carbohydrates compared to those indicated in the WHO recommendations [39,71,127,128,129]. Another limitation is the absence of specific guidelines for specific food group intakes to compare our data. Unfortunately, not only in Serbia are there no recommendations for specific food group intakes, but there is a lack of a common standard and standardized criteria for the recommended food group intake in other countries as well [130]. Recommendations are often based on different units and expressed differently, therefore not allowing us to use them in our study and perform comparative analyses [130,131].

Our study indicated several pitfalls in current vegetarian and semi-vegetarian dietary practices in Serbia. Unfortunately, in Serbia, national food system-based dietary guidelines (FSBDGs) still do not exist, not only for vegetarian dietary patterns but also for the overall adult population, and our data indicate an urgent need for their development [132]. Vegetarians and semi-vegetarians need to have more information and better education on proper dietary practices, as well as dedication to plan and prepare appropriate meals [31]. Their meals need to be better balanced, with increased consumption of plant protein sources and, in the case of non-vegans, animal protein sources, particularly eggs, fish, and reduced-fat dairy products. The guidelines should promote the increased inclusion of other plant protein sources apart from grains, considering their greater protein content and quality: to increase legume, nut, seed, and pseudo-cereal intakes (not only non-refined grains), to incorporate food fortified with extra proteins and fermented foods (to enhance protein digestibility), and to increase consumption of milk and meat alternatives but to avoid “junk (i.e., energy-dense but nutrient-poor, highly processed food) plant substitutes [31]. The guidelines should advise better combining diverse, complementary sources of proteins, better distribution of proteins throughout the day, increased consumption of ALA sources (seeds, nuts), and supplementation with omega-3 fatty acids from plant (microalgal) or fish sources. Additionally, the guidelines should advise not only the increased consumption of legumes (including soy-based products), pseudo-cereals, nuts, seeds, whole grains, vegetables, and fruit but also the decreased consumption of refined oils rich in omega-6, hydrogenated fats (margarines), non-soy-based cheese substitutes, and other industrial products rich in hydrogenated fats (including savory snacks, sweets, etc.) and the decreased consumption of items rich in refined carbohydrates, sugars (sweets, sugar-based beverages, refined grains), and alcohol. Furthermore, policies and marketing strategies should be considered to develop new plant-based products and increase consumer acceptability and affordability of healthy meat and dairy alternatives [31,36,37,133,134,135,136]. While global sustainability goals and policies promote a shift towards more environmentally friendly plant-based diets, they should also secure their appropriateness from the nutritional and health perspective [132,136,137].

## 5. Conclusions

The present study described, for the first time, vegetarian and semi-vegetarian eating patterns among individuals residing in Serbia. The study revealed multiple nutritional shortcomings concerning the macronutrient intakes in this population, indicating that each of the examined patterns could be connected with nutritional shortcomings and unhealthy practices. The most significant was an insufficient intake of proteins in the vegan population, associated with decreased protein quantity, quality, and availability in plant sources. Although protein intake insufficiency was less frequent among lacto-ovo vegetarians and semi-vegetarians, most of them still did not meet the recommended protein intake for the vegetarian population. Other nutritional shortages included a high intake of fat (particularly from sources rich in omega-6 and trans fats), which was more pronounced among the non-vegan population (but also among vegans). In all three vegetarian dietary patterns, the intake of omega-3 fatty acid sources was very low and much lower than the intake of omega-6 sources, which may lead to an imbalance between omega-3 and omega-6 intakes. Furthermore, among lacto-ovo vegetarians and semi-vegetarians, there was also a lower consumption of food rich in complex carbohydrates and fiber and an increased consumption of food rich in added sugars. The findings of this study highlight the need for improved education and information for the vegetarian and semi-vegetarian populations in Serbia, as well as the necessity for the development of national food system-based dietary guidelines for these populations.

## Figures and Tables

**Figure 1 foods-14-01285-f001:**
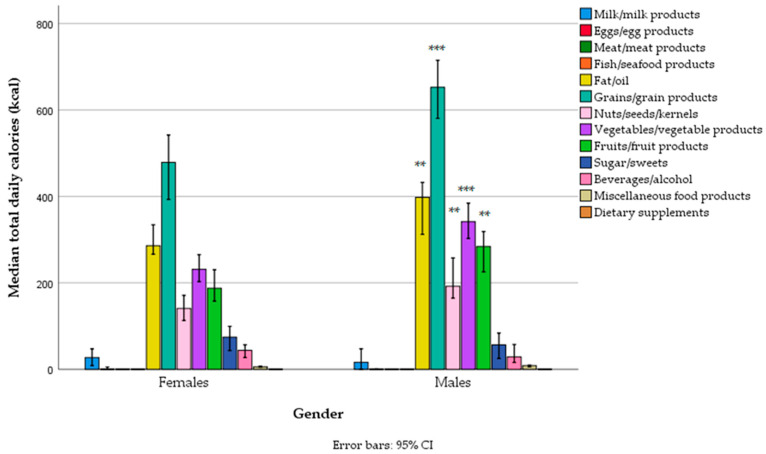
Total daily energy intake from different food groups across gender categories in a sample of vegetarians/semi-vegetarians 18–74 years old (*n* = 314) living in Serbia (** *p* < 0.01, *** *p* < 0.001).

**Figure 2 foods-14-01285-f002:**
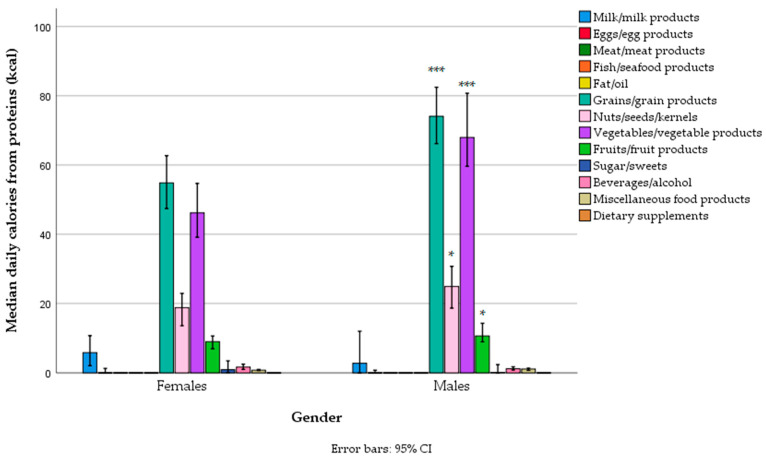
Protein-related energy intake from different food groups across gender categories in a sample of vegetarians/semi-vegetarians 18–74 years old (*n* = 314) living in Serbia (* *p* < 0.05, *** *p* < 0.001).

**Figure 3 foods-14-01285-f003:**
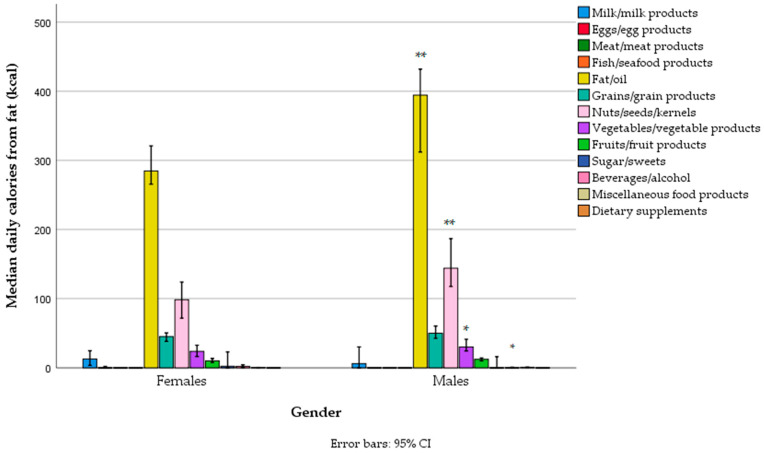
Fat-related energy intake from different food groups across gender categories in a sample of vegetarians/semi-vegetarians 18–74 years old (*n* = 314) living in Serbia (* *p* < 0.05, ** *p* < 0.01).

**Figure 4 foods-14-01285-f004:**
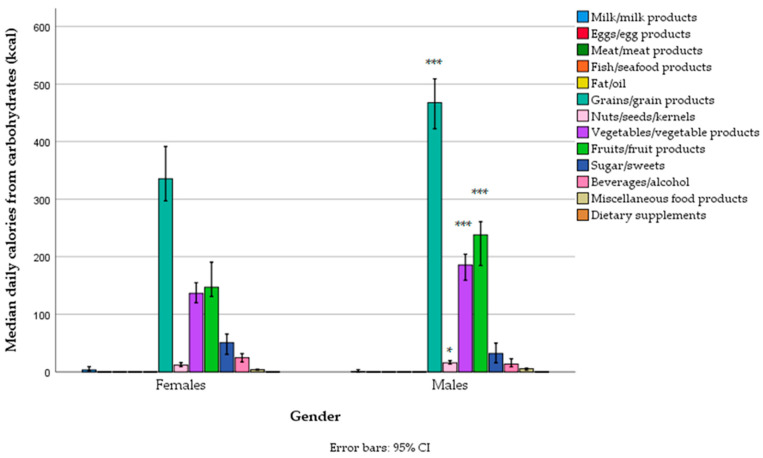
Carbohydrate-related energy intake from different food groups across gender categories in a sample of vegetarians/semi-vegetarians 18–74 years old (*n* = 314) living in Serbia (* *p* < 0.05, *** *p* < 0.001).

**Figure 5 foods-14-01285-f005:**
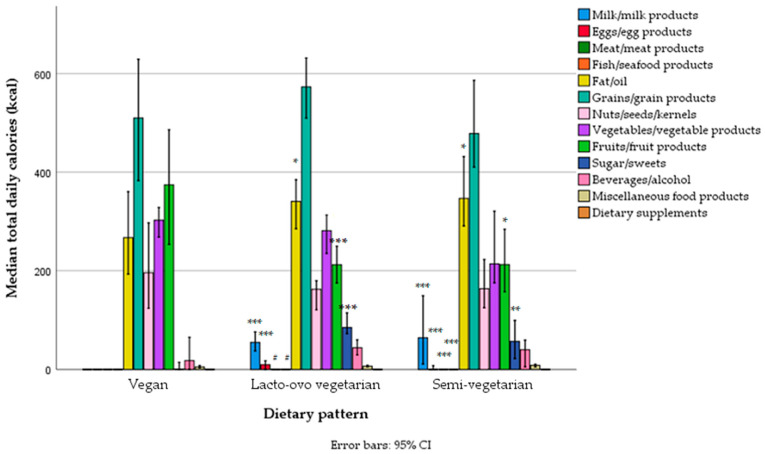
Total daily energy intake from different food groups across different vegetarian/semi-vegetarian dietary patterns in a sample of 18–64-year-old vegetarians (*n* = 314) living in Serbia. (Legend: *** *p* < 0.001, ** *p* < 0.01, * *p* < 0.001—statistically significantly different from vegans; # *p* < 0.001—statistically different from semi-vegetarians).

**Figure 6 foods-14-01285-f006:**
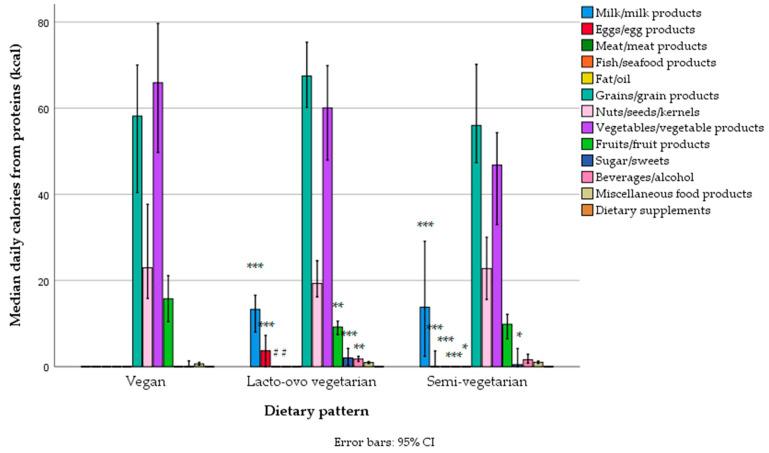
Protein-related daily energy intake from different food groups across different vegetarian/semi-vegetarian dietary patterns in a sample of 18–64-year-old vegetarians/semi-vegetarians (*n* = 314) living in Serbia. (Legend: *** *p* < 0.001, ** *p* < 0.01, * *p* < 0.001—statistically significantly different from vegans; # *p* < 0.001—statistically different from semi-vegetarians).

**Figure 7 foods-14-01285-f007:**
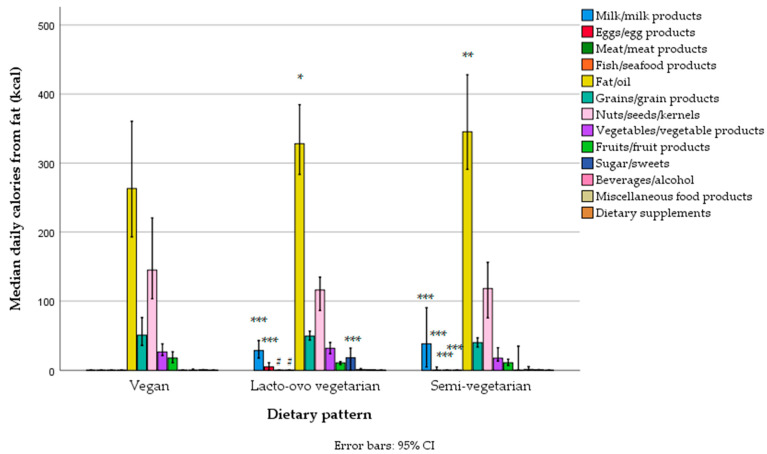
Fat-related daily energy intake from different food groups across different vegetarian/semi-vegetarian dietary patterns in a sample of 18–64-year-old vegetarians/semi-vegetarians (*n* = 314) living in Serbia. (Legend: *** *p* < 0.001, ** *p* < 0.01, * *p* < 0.001—statistically significantly different from vegans; # *p* < 0.001—statistically different from semi-vegetarians).

**Figure 8 foods-14-01285-f008:**
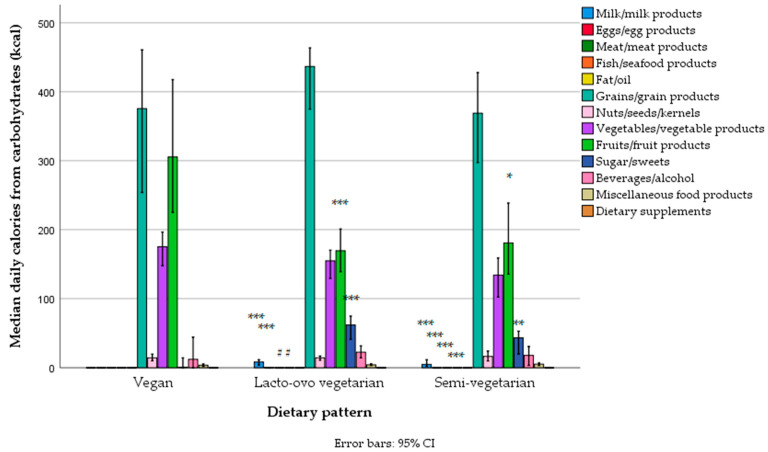
Carbohydrate-related daily energy intake from different food groups across different vegetarian/semi-vegetarian dietary patterns in a sample of 18–64-year-old vegetarians/semi-vegetarians (*n* = 314) living in Serbia. (Legend: *** *p* < 0.001, ** *p* < 0.01, * *p* < 0.001—statistically significantly different from vegans; # *p* < 0.001—statistically different from semi-vegetarians).

**Table 1 foods-14-01285-t001:** Socio-demographic characteristics of the study sample of vegetarians/semi-vegetarians living in Serbia, 18–74 years old, with gender differences (*n* = 314).

	Total Sample *n* = 314	Women *n* = 166	Men *n* =148	*p*
**Dietary pattern, *n* (%)**				0.201
Vegan	63 (20.1)	27 (16.3)	36 (24.3)	
Lacto-ovo vegetarian	192 (61.1)	107 (64.5)	85 (57.4)	
Semi-vegetarian (Pescatarian/Flexitarian)	59 (18.8)	32 (19.3)	27 (18.2)	
**Age, years, median (IQR)**	36.0 (27.0–47.0)	36.9 (26.1–48.0)	34.7 (27.7–44.0)	0.351
**Age group, *n* (%)**				0.246
18–64 years	275 (87.6)	142 (85.5)	133 (89.9)	
65–74 years	39 (12.4)	24 (14.5)	15 (10.1)	
**Distribution per geographical region, *n* (%)**				0.978
Belgrade (capital city) region	86 (27.4)	46 (27.7)	40 (27.0)	
Vojvodina region	81 (25.8)	44 (26.5)	37 (25.0)	
Region of Šumadija and Western Serbia	84 (26.8)	44 (26.5)	40 (27.0)	
South-Eastern Serbia region	63 (20.1)	32 (19.3)	31 (20.9)	
**Settlement type, *n* (%)**				0.473
Urban	290 (92.4)	155 (93.4)	135 (91.2)	
Rural	24 (7.6)	11 (6.6)	13 (8.8)	
**Ethnicity, *n* (%)**				0.058
Serbian	282 (89.8)	144 (86.7)	138 (93.2)	
Other	32 (10.2)	22 13.3)	10 (6.8)	
**Religion, *n* (%)**				0.258
Orthodox	198 (63.1)	112 (67.5)	86 (58.1)	
Catholic	10 (3.2)	4 (2.4)	6 (4.1)	
Islamic	1 (0.3)	0 (0.0)	1 (0.7)	
Other e.g., Adventist/Atheist/Agnostic	105 (33.4)	50 (30.19	55 (37.2)	
**Marital status, *n* (%)**				**0.013 ***
Single	170 (54.1)	79 (47.6)	91 (61.5)	
Married	100 (31.8)	55 (33.1)	45 (30.4)	
Divorced	18 (5.7)	15 (9.0)	3 (2.0)	
Separated	11 (3.5)	5 (3.0)	6 (4.1)	
Widowed	11 (3.5)	9 (5.4)	2 (1.4)	
Other	4 (1.3)	3 (1.8)	1 (0.7)	
**Household size, people per household, median (IQR)**	2.0 (2.0–3.0)	2.0 (2.0–3.0)	2.0 (1.0–3.0)	0.290
**Highest level of formal education, *n* (%)**				0.930
ISCED 1: Primary education	3 (1.0)	1 (0.6)	2 (1.4)	
ISCED 2: Lower secondary education	4 (1.3)	2 (1.2)	2 (1.4)	
ISCED 3: Upper secondary education	82 (26.1)	41 (24.7)	41 (27.7)	
ISCED 4/5: Post-secondary/Short-cycle tertiary education	25 (8.0)	12 (7.2)	13 (8.8)	
ISCED 6: Bachelor’s or equivalent level	142 (45.2)	78 (47.0)	64 (43.2)	
ISCED 7/8: Master’s/Doctoral or equivalent level	58 (18.5)	32 (19.3)	26 (17.6)	
**Presence of chronic illness, *n* (%)**				**0.040 ***
No	252 (80.3)	126 (75.9)	126 (85.1)	
Yes:	62 (19.7)	40 (24.1)	22 (14.9)	
Neoplasms	4 (1.3)	2 (1.2)	2 (1.4)	1.000
Diseases of the blood-forming organs and immune system	5 (1.6)	5 (3.0)	0 (0.0)	0.063
Endocrine, nutritional, and metabolic diseases	17 (5.4)	14 (8.4)	3 (2.0)	**0.013 ***
Mental and behavioral disorders	9 (2.9)	4 (2.4)	5 (3.4)	0.740
Diseases of the nervous system	6 (1.9)	4 (2.4)	2 (1.4)	0.688
Diseases of the circulatory system	18 (5.7)	12 (7.2)	6 (4.1)	0.227
Diseases of the respiratory system	6 (1.9)	4 (2.4)	2 (1.4)	0.688
Diseases of the digestive system	9 (2.9)	5 (3.0)	4 (2.7)	1.000
Diseases of the skin and subcutaneous tissue	5 (1.6)	2 (1.2)	3 (2.0)	0.669
Diseases of the musculoskeletal system and connective tissue	5 (1.6)	4 (2.4)	1 (0.7)	0.375
Diseases of the genitourinary system	6 (1.9)	4 (2.4)	2 (1.4)	0.688
Other	7 (2.2)	6 (3.6)	1 (0.7)	0.125
**Chronic medical therapy, *n* (%)**				0.194
No	282 (89.8)	145 (87.3)	137 (92.6)	
Yes	32 (10.2)	21 (12.7)	11 (7.4)	
**Smoking status, *n* (%)**				0.891
Current smoker	67 (21.3)	34 (20.5)	33 (22.3)	
Former smoker	60 (19.1)	33 (19.9)	27 (18.2)	
Never smoker	187 (59.6)	99 (59.6)	88 (59.5)	
**Physical activity level, *n* (%)**				**0.011 ***
Low	87 (27.7)	48 (28.9)	39 (26.4)	
Medium	151 (48.1)	89 (53.6)	62 (41.9)	
High	76 (24.2)	29 (17.5)	47 (31.8)	
**Physical activity METs, min/week, median (IQR)**	3492.0 (1966.5–5911.5)	3442.5 (1793.6–5602.5)	3639.0 (2085.8–7029.0)	0.220
**Season, *n* (%)**				0.829
Fall	82 (26.1)	46 (27.7)	36 (24.3)	
Spring	77 (24.5)	41 (24.7)	36 (24.3)	
Summer	76 (24.2)	37 (22.3)	39 (26.4)	
Winter	79 (25.2)	42 (25.3)	37 (25.0)	

ISCED—International Standard Classification of Education; METs—Metabolic equivalents of the task; IQR—interquartile range; *p*—statistical significance of difference (bolded values are statistically significant, * *p* < 0.05). Differences between men and women were tested with Mann–Whitney test (for numerical data) and with Chi-square test or Fisher’s Exact test (for categorical data).

**Table 2 foods-14-01285-t002:** Anthropometric data and nutritional status based on body mass index (BMI) categories in a sample of vegetarians/semi-vegetarians living in Serbia, 18–74 years old, with gender differences (*n* = 314).

	Total (*n* = 314)	Women (*n* = 166)	Men (*n* = 148)	*p*
	Median (IQR)	Median (IQR)	Median (IQR)	
**Body height (cm)**	174.0 (167.0–181.0)	168.0 (163.0–173.0)	181.0 (177.0–185.0)	**<0.001 *****
**Body mass (kg)**	68.0 (60.0–78.0)	61.0 (56.0–67.0)	78.0 (71.3–84.0)	**<0.001 *****
**BMI (kg/m^2^)**	22.6 (20.5–24.7)	21.5 (19.9–23.7)	23.8 (22.2–25.4)	**<0.001 *****
**BMI categories, n (%)**				**<0.001 *****
Underweight	17 (5.4)	14 (8.4)	3 (2.0)	
Normal weight	227 (72.3)	128 (77.1)	99 (66.9)	
Overweight	61 (19.4)	22 (13.3)	39 (26.4)	
Obese	9 (2.9)	2 (1.2)	7 (4.7)	

BMI—body mass index. IQR—interquartile range; *p*—statistical significance of difference (bolded values are statistically significant, *** *p* < 0.001). Differences between men and women were tested with Mann–Whitney test (for numerical data) and with the Chi-square test or Fisher’s exact test (for categorical data).

**Table 3 foods-14-01285-t003:** Daily energy and macronutrient intakes in a sample of vegetarians/semi-vegetarians living in Serbia, 18–74 years old, with gender differences (*n* = 314).

	Total (*n* = 314)	Women (*n* = 166)	Men (*n* = 148)	*p*
	Median (IQR)	Median (IQR)	Median (IQR)	
**Total energy (kcal)**	2053.5 (1663.0–2548.8)	1831.0 (1495.5–2154.5)	2465.5 (1982.8–3022.3)	**<0.001 *****
**Total energy (kcal/kg body mass)**	30.1 (25.0–37.6)	29.6 (23.6–36.2)	31.2 (25.7–38.3)	0.058
**Protein (g)**	57.3 (40.8–73.0)	48.3 (36.5–61.9)	67.4 (51.1–86.3)	**<0.001 *****
**Protein (g/kg body mass)**	0.81 (0.61–1.08)	0.75 (0.59–1.07)	0.87 (0.63–1.09)	0.137
**Protein intake adequacy (general population, AR, EAR, 50%) ^1,2^, *n* (%)**				0.110
<0.66 g/kg body mass (insufficient)	101 (32.2)	60 (36.1)	41 (27.7)	
≥0.66 g/kg body mass (sufficient)	213 (67.8)	106 (63.9)	107 (72.3)	
**Protein intake adequacy (general population, PRI, 97.5%) ^1^, *n* (%)**				**0.020 ***
<0.83 g/kg body mass (insufficient)	164 (52.2)	97 (58.4)	67 (45.3)	
≥0.83 g/kg body mass (sufficient)	150 (48.7)	69 (41.6)	81 (54.7)	
**Protein intake categories (general population, RDA, 97.5%) ^2^, *n* (%)**				**0.011 ***
<0.80 g/kg body mass (insufficient)	151 (48.1)	91 (54.8)	60 (40.5)	
≥0.80 g/kg body mass (sufficient)	163 (51.9)	75 (45.2)	88 (59.5)	
**Protein intake adequacy (vegetarian population), *n* (%)**				0.872
<1.00 g/kg body mass (insufficient)	215 (68.5)	113 (68.1)	102 (68.9)	
≥1.00 g/kg body mass (sufficient)	99 (31.5)	53 (31.9)	46 (31.1)	
**Fat (g)**	82.6 (61.6–113.6)	73.2 (56.4–98.2)	98.9 (72.6–125.1)	**<0.001 *****
**Carbohydrates (g)**	246.3 (185.7–301.1)	209.6 (171.9–262.9)	283.4 (227.4–348.8)	**<0.001 *****
**Dietary fiber (g)**	34.6 (27.2–45.7)	32.1 (22.5–39.9)	40.8 (31.2–54.9)	**<0.001 *****
**Dietary fiber intake adequacy (AI) ^1^, *n* (%)**				**<0.001 *****
<25 g (insufficient)	68 (21.7)	53 (31.9)	15 (10.1)	
≥25 g (sufficient)	246 (78.3)	113 (68.1)	133 (89.9)	
**Dietary fiber intake adequacy (AI) ^2^, *n* (%)**				0.874
<14 g/1000 kcal (insufficient)	92 (29.3)	48 (28.9)	44 (29.7)	
≥14 g/1000 kcal (sufficient)	222 (78.3)	118 (71.8%)	104 (70.3)	
**Alcohol (g)**	0.0 (0.0–0.1)	0.0 (0.0–0.1)	0.0 (0.0–0.1)	0.225
**Protein (kcal)**	229.3 (163.2–292.1)	193.4 (146.1–247.6)	269.4 (204.4–345.1)	**<0.001 *****
**Fat (kcal)**	743.0 (554.0–1022.2)	658.9 (507.8–883.4)	890.0 (653.0–1126.2)	**<0.001 *****
**Carbohydrates (kcal)**	985.1 (742.8–1204.5)	838.3 (687.7–1051.6)	1133.8 (909.4–1395.2)	**<0.001 *****
**Dietary fiber (kcal)**	69.2 (54.5–91.5)	64.1 (45.1–79.8)	81.6 (62.3–109.8)	**<0.001 *****
**Alcohol (kcal)**	0.0 (0.0–0.6)	0.0 (0.0–0.4)	0.1 (0.0–1.0)	0.225
**Protein (%TE)**	10.9 (9.2–12.4)	10.9 (9.2–12.2)	10.9 (9.1–12.8)	0.970
**Fat (%TE)**	37.1 (31.1–42.6)	37.8 (32.6–41.9)	36.3 (29.5–43.6)	0.220
**Carbohydrates (%TE)**	46.7 (41.5–53.4)	46.7 (41.9–52.0)	47.1 (40.7–54.7)	0.752
**Dietary fiber (%TE)**	0.0 (0.0–0.0)	0.0 (0.0–0.0)	0.0 (0.0–0.0)	0.821
**Alcohol (%TE)**	0.0 (0.0–0.0)	0.0 (0.0–0.0)	0.0 (0.0–0.0)	0.481

^1^ EFSA (European Food Safety Authority). Dietary Reference Values for Nutrients—Summary Report. EFSA supporting publications. December 2017. Available at https://efsa.onlinelibrary.wiley.com/ (accessed on 20 January 2025). ^2^ US Department of Agriculture and US Department of Health and Human Services. Dietary Guidelines for Americans, 2020–2025. 9th Edition. December 2020. Available at https://www.dietaryguidelines.gov/ (accessed on 20 January 2025). AR—average requirement; PRI—Population Reference Intake; EAR—Estimated Average Requirement; RDA—Recommended Dietary Allowance; AI—Adequate Intake; %TE—percentage of total energy; IQR—interquartile range; *p*—statistical significance of difference (bolded values are statistically significant, * *p* < 0.05, *** *p* < 0.001). Differences between men and women were tested with the Mann–Whitney test (for numerical data) and with the Chi-square test or Fisher’s exact test (for categorical data).

**Table 4 foods-14-01285-t004:** Comparison of categories in energy intake and proportion of energy from macronutrients with the EFSA ^1^ and the US ^2^ dietary guidelines.

	Total (*n* = 314)	Women (*n* = 166)	Men (*n* = 148)	
	*n* (%)	*n* (%)	*n* (%)	*p*
**Total energy intake ^1,2^:**				0.187
<1600 kcal/day (women), <2000 kcal/day (men)	92 (29.3)	54 (32.5)	38 (25.7)	
1600–2400 kcal/day (women), 2000–3000 kcal/day (men)	154 (49.0)	82 (49.4)	53 (35.8)	
≥2400 kcal/day (women), ≥3000 kcal/day (men)	68 (21.7)	30 (18.1)	38 (25.7)	
**Percentage of energy intake coming from proteins ^2^:**				0.430
<10% (insufficient)	124 (39.5)	69 (41.6)	55 (37.2)	
10–35% (adequate)	189 (60.2)	97 (58.4)	92 (62.2)	
>35% (excessive)	1 (0.3)	0 (0.0)	1 (0.7)	
**Percentage of energy intake coming from fats ^1,2^:**				**0.021 ***
<20% (insufficient)	14 (4.5)	3 (1.8)	11 (7.4)	
20–35% (adequate)	112 (35.7)	55 (33.1)	57 (38.5)	
>35% (excessive)	188 (59.9)	108 (65.1)	80 (54.1)	
**Percentage of energy intake coming from carbohydrates ^1^:**				0.078
<45% (insufficient)	127 (40.4)	64 (38.6)	63 (42.6)	
45–60% (adequate)	157 (50.0)	91 (54.8)	66 (44.6)	
>60% (excessive)	30 (9.6)	11 (6.6)	19 (12.8)	
**Percentage of energy intake coming from carbohydrates ^2^:**				0.215
<45% (insufficient)	127 (40.4)	64 (38.6)	63 (42.6)	
45–65% (adequate)	177 (56.4)	99 (59.6)	78 (52.7)	
>65% (excessive)	10 (3.2)	3 (1.8)	7 (4.7)	

^1^ EFSA (European Food Safety Authority). Dietary Reference Values for Nutrients—Summary Report. EFSA supporting publications. December 2017. Available at https://efsa.onlinelibrary.wiley.com/ (accessed on 20 January 2025). ^2^ US Department of Agriculture and US Department of Health and Human Services. Dietary Guidelines for Americans, 2020–2025. 9th Edition. December 2020. Available at https://www.dietaryguidelines.gov (accessed on 20 January 2025). *p*—statistical significance of difference (bolded values are statistically significant, * *p* < 0.05). Differences between men and women were tested with the Chi-square test or Fisher’s exact test.

**Table 5 foods-14-01285-t005:** Contribution of food groups to total energy intake in a sample of vegetarians/semi-vegetarians living in Serbia, 18–74 years old, with gender differences (*n* = 314).

Food Group	Total (*n* = 314)	Women (*n* = 166)	Men (*n* = 148)	*p*
	Median (IQR)	Median (IQR)	Median (IQR)	
**Milk/milk products (%TE)**	1.2 (0.0–8.1)	1.6 (0.0–9.4)	0.6 (0.0–5.9)	0.183
**Eggs/egg products %TE**	0.0 (0.0–1.2)	0.0 (0.0–1.3)	0.0 (0.0–1.2)	0.898
**Meat/meat products (%TE)**	0.0 (0.0–0.0)	0.0 (0.0–0.0)	0.0 (0.0–0.0)	0.880
**Fish/seafood products (%TE)**	0.0 (0.0–0.0)	0.0 (0.0–0.0)	0.0 (0.0–0.0)	0.408
**Fat/oil (%TE)**	16.0 (11.2–22.7)	17.3 (12.4–22.5)	15.4 (10.3–22.9)	0.077
**Grains/grain products (%TE)**	26.4 (18.2–33.9)	26.9 (19.2–33.8)	26.0 (16.5–34.0)	0.621
**Nuts/seeds/kernels (%TE)**	8.1 (2.5–15.7)	8.1 (2.3–14.6)	8.1 (2.6–16.7)	0.591
**Vegetables/vegetable products (%TE)**	13.0 (8.1–19.8)	12.5 (8.4–19.0)	13.0 (8.0–20.2)	0.892
**Fruits/fruit products (%TE)**	11.6 (5.9–18.8)	11.4 (5.9–18.6)	11.8 (5.9–20.6)	0.783
**Sugar/sweets (%TE)**	2.8 (0.1–8.5)	3.6 (0.5–9.9)	2.1 (0.0–8.0)	**0.030 ***
**Beverages/alcohol (%TE)**	1.8 (0.2–6.3)	2.1 (0.3–6.3)	1.2 (0.1–6.3)	0.303
**Miscellaneous food products (%TE)**	0.3 (0.2–0.6)	0.3 (0.2–0.6)	0.3 (0.1–0.6)	0.340
**Dietary supplements (%TE)**	0.0 (0.0–0.0)	0.0 (0.0–0.0)	0.0 (0.0–0.0)	0.890

%TE—percentage of total energy; IQR—interquartile range; *p*—statistical significance of difference (bolded values are statistically significant, * *p* < 0.05). Differences between men and women were tested with the Mann–Whitney test.

**Table 6 foods-14-01285-t006:** Differences in socio-demographic and anthropometric data between different vegetarian dietary patterns in a sample of vegetarians/semi-vegetarians living in Serbia, 18–74 years old, with gender differences (*n* = 314).

	Vegan (*n* = 63)	Lacto-Ovo Vegetarian (*n* =192)	Semi-Vegetarian (*n* = 59)	*p*	Post Hoc *p*		
	Median (IQR)	Median (IQR)	Median (IQR)		V vs. LOV	V vs. SV	LOV vs. SV
**Age (years)**	38.0 (31.0–50.7)	33.6 (25.0–44.8)	41.7 (29.0–55.0)	**<0.001 *****	**0.018 ***	1.000	**0.003 ****
**Distribution per geographical region, *n* (%)**				**0.001 ****	0.283	**0.003 ***	0.086
Belgrade region	26 (41.3)	47 (24.5)	13 (22.0)				
Vojvodina region	10 (15.9)	51 (26.6)	20 (33.9)				
Šumadija&Western S.	19 (30.2)	58 (30.2)	7 (11.9)				
South-Eastern Serbia	8 (12.7)	36 (18.8)	19 (32.2)				
**Religion, *n* (%)**				**0.014 ***	0.071	0.990	1.000
Orthodox	37 (58.7)	120 (62.5)	41 (69.5)				
Catholic	6 (9.5)	3 (1.6)	1 (1.7)				
Islamic	0 (0.0)	0 (0.0)	1 (1.7)				
Other	20 (31.7)	69 (35.9)	16 (27.1)				
**Smoking status, *n* (%)**				**0.029 ***	0.214	1.000	1.000
Current smoker	8 (12.7)	47 (24.5)	12 (20.3)				
Former smoker	16 (25.4)	27 (14.1)	17 (28.8)				
Never smoker	39 (61.9)	118 (61.5)	30 (50.8)				
**Physical activity level, *n* (%)**				0.059			
Low	13 (20.6)	58 (30.2)	16 (27.1)				
Medium	26 (41.3)	93 (48.4)	32 (54.2)				
High	24 (38.1)	41 (21.4)	11 (18.6)				
**Physical activity METs, min/week**	4293.0 (1908.0–7812.0)	3492.5 (1924.5–5529.0)	3066.0 (1986.0–4572.0)	0.178			
**Body height (cm)**	178.0 (169.0–183.0)	174.5 (167.3–180.0)	172.0 (165.0–180.0)	0.151			
**Body mass (kg)**	67.0 (59.0–80.0)	70.0 (60.0–78.0)	68.0 (60.0–79.0)	0.992			
**BMI (kg/m^2^)**	22.4 (20.3–24.0)	22.6 (20.6–24.7)	22.7 (21.1–25.2)	0.355			
**BMI categories, *n* (%)**				0.120			
Underweight	6 (9.5)	11 (5.7)	0 (0.0)				
Normal weight	44 (69.8)	140 (72.9)	43 (72.9)				
Overweight	13 (20.6)	33 (17.2)	15 (25.4)				
Obese	0 (0.0)	8 (4.2)	1 (1.7)				

BMI—body mass index; METs– Metabolic equivalents of the task; V—vegan; LOV—lacto-ovo vegetarian; SV—semi-vegetarian; IQR—interquartile range; *p*—statistical significance of difference (bolded values are statistically significant, * *p* < 0.05, ** *p* < 0.01, *** *p* < 0.001). Differences between different dietary patterns were tested with the Kruskal–Wallis test with the post hoc Dunn’s test adjusted by the Bonferroni correction for multiple tests (for numerical data) and with the Chi-square test or Fisher’s exact test (with the post hoc Z-test adjusted by the Bonferroni correction for multiple tests (for categorical data).

**Table 7 foods-14-01285-t007:** Differences in daily energy and macronutrient intake between different vegetarian/semi-vegetarian dietary patterns in a sample of vegetarians/semi-vegetarians living in Serbia, 18–74 years old, with gender differences (*n* = 314).

	Vegan (*n* = 63)	Lacto-Ovo Vegetarian (*n* =192)	Semi-Vegetarian (*n* = 59)	*p*	Post Hoc *p*		
	Median (IQR)	Median (IQR)	Median (IQR)		V vs. LOV	V vs. SV	LOV vs. SV
**Total energy (kcal)**	2043.0 (1618.0–2479.0)	2093.0 (1679.3–2593.5)	2015.0 (1654.0–2637.0)	0.521			
**Total energy** **(kcal/kg body mass)**	29.7 (23.9–37.1)	30.2 (24.9–37.6)	29.2 (25.2–37.9)	0.649			
**Protein (g)**	46.5 (32.2–72.1)	58.9 (44.1–73.0)	59.3 (39.5–76.4)	**0.018 ***	**0.014 ***	0.149	1.000
**Protein (g/kg body mass)**	0.69 (0.50–0.98)	0.83 (0.64–1.10)	0.85 (0.62–1.04)	**0.015 ***	**0.011 ***	0.228	1.000
**Protein intake adequacy (general population AR, EAR, 50%) ^1,2^, *n* (%)**							
<0.66 g/kg body mass	29 (46.0)	54 (28.1)	18 (30.5)	**0.029 ***	**0.024 ***	0.234	1.000
≥0.66 g/kg body mass	34 (54.0)	138 (71.9)	41 (69.5)				
**Protein intake adequacy (general population PRI, 97.5%) ^1^, *n* (%)**							
<0.83 g/kg body mass	41 (65.1)	96 (50.0)	27 (45.8)	0.063			
≥0.83 g/kg body mass	22 (34.9)	96 (50.0)	32 (54.2)				
**Protein intake adequacy (general population, RDA, 97,5%) ^2^, *n* (%)**							
<0.8 g/kg body mass	39 (61.9)	86 (44.8)	26 (44.1)	**0.049 ***	0.055	0.145	1.000
≥0.8 g/kg body mass	24 (38.1)	106 (55.2)	33 (55.9)				
**Protein intake adequacy (vegetarian population), *n* (%)**							
<1.00 g/kg body mass	48 (76.2)	124 (64.6)	43 (72.9)	0.146			
≥1.00 g/kg body mass	15 (23.8)	68 (35.4)	16 (27.1)				
**Fat (g)**	68.5 (48.5–98.4)	84.4 (64.5–117.7)	83.5 (64.0–110.8)	**0.002 ****	**0.002 ****	**0.030 ***	1.000
**Carbohydrates (g)**	259.0 (206.1–338.9)	248.7 (182.0–298.9)	217.2 (181.7–275.6)	0.089			
**Dietary fiber (g)**	40.8 (30.2–55.8)	32.9 (26.4–43.3)	35.5 (27.4–43.1)	**0.006 ****	**0.004 ****	0.112	1.000
**Dietary fiber intake categories (AI)** **^1^, *n* (%)**							
<25 g (insufficient)	9 (14.3)	45 (23.4)	14 (23.7)	0.283			
≥25 g (sufficient)	54 (85.7)	147 (76.6)	45 (76.3)				
**Dietary fiber intake categories (AI) ^2^, *n* (%)**							
<14 g/1000 kcal (insuff.)	8 (12.7%)	63 (32.8%)	21 (35.6%)	**0.005 ****	**0.006 ****	**0.009 ****	1.000
≥14 g/1000 kcal (suff.)	55 (87.3)	129 (67.2)	38 (64.4%)				
**Alcohol (g)**	0.0 (0.0–0.1)	0.0 (0.0–0.1)	0.0 (0.0–0.1)	0.390			
**Protein (kcal)**	185.9 (128.9–288.4)	235.6 (176.5–292.1)	237.3 (158.0–305.5)	**0.018 ***	**0.014 ***	0.149	1.000
**Fat (kcal)**	616.1 (436.7–885.6)	759.7 (580.6–1059.5)	751.8 (575.8–997.4)	**0.002 ****	**0.002 ****	**0.030 ***	1.000
**Carbohydrates (kcal)**	1035.9 (824.5–1355.5)	994.8 (727.9–1195.4)	868.8 (726.7–1102.3)	0.089			
**Dietary fiber (kcal)**	81.6 (60.5–111.5)	65.9(52.8–86.5)	71.0 (54.9–86.1)	**0.006 ****	**0.004 ****	0.112	1.000
**Alcohol (kcal)**	0.0 (0.0–0.5)	0.1 (0.0–1.1)	0.0 (0.0–0.4)	0.390			
**Protein (%TE)**	9.2 (7.1–12.2)	11.1 (9.5–12.4)	10.9 (9.5–12.5)	**0.002 ****	**0.001 ****	**0.043 ***	1.000
**Fat (%TE)**	32.1 (26.1–38.5)	37.5 (32.1–43.3)	40.1 (33.2–44.5)	**<0.001 *****	**<0.001 *****	**<0.001 *****	0.921
**Carbohydrates (%TE)**	51.7(45.1–60.4)	46.2 (41.7–51.2)	44.0 (38.6–51.0)	**<0.001 *****	**<0.001 *****	**<0.001 *****	0.527
**Dietary fiber (%TE)**	0.0 (0.0–0.1)	0.0 (0.00.0)	0.0 (0.0–0.0)	**<0.001 *****	**<0.001 *****	**0.003 ****	0.928
**Alcohol (%TE)**	0.0 (0.0–0.0)	0.0 (0.0–0.1)	0.0 (0.0–0.0)	0.518			

^1^ EFSA (European Food Safety Authority). Dietary Reference Values for Nutrients—Summary Report. EFSA supporting publications. December 2017. Available at https://efsa.onlinelibrary.wiley.com/ (accessed on 20 January 2025). ^2^ US Department of Agriculture and US Department of Health and Human Services. Dietary Guidelines for Americans, 2020–2025. 9th Edition. December 2020. Available at https://www.dietaryguidelines.gov/ (accessed on 20 January 2025). AR—average requirements; PRI—Population Reference Intakes; EAR—Estimated Average Requirements; RDA—Recommended Dietary Allowances; AI—Adequate intakes; %TE—percentage of total energy; V—vegan; LOV—lacto-ovo vegetarian; SV—semi-vegetarian; IQR—interquartile range; *p*—statistical significance of difference (bolded values are statistically significant, * *p* < 0.05, ** *p* < 0.01, *** *p* < 0.001). Differences between different dietary patterns were tested with the Kruskal–Wallis test with the post hoc Dunn’s test adjusted by the Bonferroni correction for multiple tests (for numerical data) and with the Chi-square test or Fisher’s exact test (with the post hoc Z-test adjusted by the Bonferroni correction for multiple tests (for categorical data).

**Table 8 foods-14-01285-t008:** Different vegetarian/semi-vegetarian dietary patterns in comparison with the EFSA ^1^ and US ^2^ dietary guidelines for energy intake and proportion of energy from macronutrients.

	Vegan (*n* = 63)	Lacto-Ovo Vegetarian (*n* =192)	Semi-Vegetarian (*n* = 59)	*p*	Post Hoc *p*		
	n (%)	n (%)	n (%)		V vs. LOV	V vs. SV	LOV vs. SV
**Total energy intake ^1,2^:**				0.532			
<1600 kcal/day (W), <2000 kcal/day (M)	22 (34.9)	52 (27.1)	18 (30.5)				
1600–2400 kcal/day (W), 2000–3000 kcal/day (M)	32 (50.8)	94 (49.0)	28 (47.5)				
≥2400 kcal/day (W), ≥3000 kcal/day (M)	9 (14.3)	46 (24.0)	13 (22.0)				
**Percentage of energy intake coming from proteins ^2^:**				**0.002 ***	**<0.001 *****	**0.011 ***	1.000
<10% (insufficient)	39 (61.9)	64 (33.3)	21 (35.6)				
10–35% (adequate)	24 (38.1)	127 (66.1)	38 (64.4)				
>35% (excessive)	0 (0.0)	1 (0.5)	0 (0.0)				
**Percentage of energy intake coming from fats ^1,2^:**				**<0.001 *****	**<0.001 *****	**0.006 ****	1.000
<20% (insufficient)	10 (15.9)	2 (1.0)	2 (3.4)				
20–35% (adequate)	27 (42.9)	70 (36.5)	15 (25.4)				
>35% (excessive)	26 (41.3)	120 (62.5)	42 (71.2)				
**Percentage of energy intake coming from carbohydrates ^1^:**				**<0.001 *****	**<0.001 *****	**0.001 ****	0.151
<45% (insufficient)	14 (22.2)	82 (42.7)	31 (52.5)				
45–60% (adequate)	30 (47.6)	104 (54.2)	23 (39.0)				
>60% (excessive)	19 (30.2)	6 (3.1)	5 (8.5)				
**Percentage of energy intake coming from carbohydrates ^2^:**				**<0.001 *****	**<0.001 *****	**0.002 ****	1.000
<45% (insufficient)	14 (22.2)	82 (42.7)	31 (52.5)				
45–65% (adequate)	41 (65.1)	109 (56.8)	27 (45.8)				
>65% (excessive)	8 (12.7)	1 (0.5)	1 (1.7)				

^1^ EFSA (European Food Safety Authority). Dietary Reference Values for Nutrients—Summary Report. EFSA supporting publications. December 2017. Available at https://efsa.onlinelibrary.wiley.com/ (accessed on 20 January 2025). ^2^ US Department of Agriculture and US Department of Health and Human Services. Dietary Guidelines for Americans, 2020–2025. 9th Edition. December 2020. Available at https://www.dietaryguidelines.gov/ (accessed on 20 January 2025). V—vegan; LOV—lacto-ovo vegetarian; SV—semi-vegetarian; W—women; M—men; IQR—interquartile range; *p*—the statistical significance of difference (bolded values are statistically significant, * *p* < 0.05, ** *p* < 0.01, *** *p* < 0.001). Differences between different dietary patterns were tested with the Kruskal–Wallis test with the post hoc Dunn’s test adjusted by the Bonferroni correction for multiple tests (for numerical data) and with the Chi-square test or Fisher’s exact test (with the post hoc Z-test adjusted by the Bonferroni correction for multiple tests (for categorical data).

**Table 9 foods-14-01285-t009:** Comparison between different vegetarian/semi-vegetarian dietary patterns in the contribution of specific food groups to total energy intake in a sample of vegetarians/semi-vegetarians 18–74 years old, living in Serbia (*n* = 314).

Food Group	Vegan (*n* = 63)	Lacto-Ovo Vegetarian (*n* = 192)	Semi-Vegetarian (*n* = 59)	*p*	Post Hoc *p*		
	Median (IQR)	Median (IQR)	Median (IQR)		V vs. LOV	V vs. SV	LOV vs. SV
**Milk/milk products (%TE)**	0.0 (0.0–0.0)	2.9 (0.0–9.8)	3.0 (0.0–10.9)	**<0.001 *****	**<0.001 *****	**<0.001 *****	1.000
**Eggs/egg products %TE**	0.0 (0.0–0.0)	0.5 (0.0–2.0)	0.0 (0.0–1.6)	**<0.001 *****	**<0.001 *****	**<0.001 *****	0.588
**Meat/meat products (%TE)**	0.0 (0.0–0.0)	0.0 (0.0–0.0)	0.0 (0.0–0.0)	**<0.001 *****	1.000	**<0.001 *****	**<0.001 *****
**Fish/seafood products (%TE)**	0.0 (0.0–0.0)	0.0 (0.0–0.0)	0.0 (0.0–1.6)	**<0.001 *****	1.000	**<0.001 *****	**<0.001 *****
**Fat/oil (%TE)**	13.2 (7.3–21.2)	15.9 (11.7–22.7)	18.0 (12.8–23.7)	**0.005 ****	**0.028 ***	**0.006 ****	0.630
**Grains/grain products (%TE)**	23.2 (10.3–35.6)	27.5 (19.4–34.3)	25.0 (18.9–30.5)	0.124			
**Nuts/seeds/kernels (%TE)**	10.6 (2.2–18.1)	7.4 (2.7–13.5)	8.9 (2.1–13.4)	0.321			
**Vegetables/vegetable products (%TE)**	15.8 (8.7–22.9)	12.6 (8.3–19.4)	10.9 (6.9–17.6)	**0.020 ***	0.161	**0.017 ***	0.409
**Fruits/fruit products (%TE)**	18.1 (7.1–38.5)	10.7 (5.8–16.3)	11.5 (4.5–20.1)	**<0.001 *****	**<0.001 *****	**0.034 ***	0.867
**Sugar/sweets (%TE)**	0.0 (0.0–4.6)	3.8 (0.8–10.6)	2.4 (0.4–7.3)	**<0.001 *****	**<0.001 *****	**0.010 ***	0.340
**Beverages/alcohol (%TE)**	0.9 (0.0–6.5)	2.1 (0.3–6.5)	1.9 (0.2–6.0)	0.136			
**Miscellaneous food products (%TE)**	0.3 (0.1–0.6)	0.3 (0.2–0.5)	0.4 (0.2–0.6)	0.200			
**Dietary supplements (%TE)**	0.0 (0.0–0.0)	0.0 (0.0–0.0)	0.0 (0.0–0.0)	0.176			

%TE—percentage of total energy; V—vegan; LOV—lacto-ovo vegetarian; SV—semi-vegetarian; IQR—interquartile range; *p*—statistical significance of difference (bolded values are statistically significant, * *p* < 0.05, ** *p* < 0.01, *** *p* < 0.001). Differences between different dietary patterns were tested with the Kruskal–Wallis test with the post hoc Dunn’s test adjusted by the Bonferroni correction for multiple tests.

## Data Availability

The data are not publicly available due to the reports using the same set of data for analysis has not yet been published. Further inquiries can be directed to the corresponding author.

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
