# Peer review of "Energy and Macronutrient Dietary Intakes of Vegetarian and Semi-Vegetarian Serbian Adults: Data from the EFSA EU Menu Food Consumption Survey (2017–2022)"

_foods, 2025, doi:10.3390/foods14081285_

Round 1
Reviewer 1 Report
Comments and Suggestions for Authors
It is a pleasure to review the article entitled "Adequacy of energy and macronutrient dietary intakes and nutritional status of vegetarian and semi-vegetarian adult population in Serbia: data from the EFSA EU Menu Food Consumption Survey in Serbia (2017-2022)". In this paper, authors examine dietary patterns and nutritional adequacy among Serbian vegetarians and semi-vegetarians through analysis of survey data from 314 participants. The study reveals concerning nutritional gaps, particularly insufficient protein intake among vegans and high fat consumption across groups. Furthermore, the paper identifies distinct dietary patterns between vegan and non-vegan populations in terms of food group contributions to energy and macronutrient intakes. These findings are timely and valuable for developing targeted dietary guidelines for plant-based populations in Serbia. That being said, there are several weaknesses that need to be addressed, and are listed below:
• The study's sample size (n=314) is relatively small for a nationally representative survey, especially given the diversity of dietary patterns (vegans, lacto-ovo vegetarians, pescatarians, flexitarians) and the stratification across age, gender, and geographical regions. The authors should justify how this sample adequately reflects the broader Serbian vegetarian and semi-vegetarian population.
• While the manuscript highlights insufficient protein intake as a key finding, it lacks detailed data on essential amino acid profiles or biological markers (e.g., serum albumin, nitrogen balance) to substantiate claims about protein quality and adequacy. Including such measures would strengthen the conclusions.
• The reliance on two 24-hour dietary recalls introduces potential recall bias and may not capture long-term dietary habits. The authors should acknowledge this limitation more explicitly and discuss how it might affect the reliability of their macronutrient intake estimates.
• The comparison of dietary intakes to EFSA and US guidelines is reasonable given the absence of Serbian national guidelines, but the manuscript does not sufficiently explain why these specific standards were chosen over others (e.g., WHO recommendations) or how applicable they are to the Serbian context.
• Table 1 provides a comprehensive overview of socio-demographic characteristics, yet the statistical analysis section does not clarify whether these variables (e.g., education, physical activity, chronic diseases) were adjusted for in the dietary intake analyses. This omission weakens the interpretation of gender and dietary pattern differences.
• The discussion of omega-3 fatty acid deficiencies is compelling, but the manuscript would benefit from quantifying the intake of alpha-linolenic acid (ALA) and its estimated conversion rates to EPA/DHA in this population. Without this, the argument about omega-3 inadequacy remains speculative.
• Although the study identifies high fat intake (particularly omega-6 and trans fats) as a concern, it does not provide a breakdown of specific food sources contributing to this pattern beyond general categories (e.g., sunflower oil, margarine). More granular data would enhance the practical utility of the findings.
• The authors claim that the vegetarian and semi-vegetarian diets in Serbia differ significantly from those in Western countries, yet the comparison is based on a limited selection of studies (e.g., EPIC-Oxford, Nutrinet-Santé). Expanding the scope of international comparisons or providing a meta-analytic context would bolster this assertion.
• The manuscript’s call for national food system-based dietary guidelines is well-supported by the data, but it lacks a clear framework or specific recommendations for what these guidelines should entail beyond broad suggestions (e.g., “better mixing of protein sources”). Offering actionable examples would increase the study’s impact.
Reviewer 2 Report
Comments and Suggestions for Authors
The research is extremely interesting and useful, as in general there is little reliable research on the nutrition and dietary intake of vegans and vegetarians. The authors did a great job and deserve recognition.
I didn't find anything serious to criticize in the manuscript in the present form.
But the manuscript is extremely long, and after a while the reader loses the thread and loses interest. It was almost impossible to read the manuscript in one go, it's so long.
If the reader does not read the entire paper, he will not get important information. 8 figures, 8 tables, 14 supplementary tables, this is too much information to appear in a single article. There are also a lot of references, more than 120.
I recommend that the authors divide the manuscript into at least two parts and report the research results in two papers.
Reviewer 3 Report
Comments and Suggestions for Authors
The article "Adequacy of energy and macronutrient dietary intakes and nutritional status of vegetarian and semi-vegetarian adult population in Serbia: data from the EFSA EU Menu Food Consumption Survey in Serbia (2017-2022)" is well-written, although it is lengthy and could be more effective if some non-essential information were eliminated and it focused more on the key points.
- The title is too long and complex; I suggest shortening it and making it more concise, while also adding some keywords to better describe the article.
- The introduction could be more concise. I recommend summarizing the key points of the study to highlight its purpose.
- The results section is too long and could be better organized. I recommend using subheadings to divide the different sections.
- The discussion section is too long and could be more focused. It is recommended to pay more attention to the implications of the results and to avoid repeating information already presented in the results section.
- The conclusions section, on the other hand, is brief. It is recommended to expand it, summarize the key points of the study, and highlight the most significant results.
- The references are complete and up-to-date.
In addition to the structural and formatting indications of the article, I express my considerations regarding the content:
It is a cross-sectional study conducted on a sample of Serbian vegetarian and semi-vegetarian adults. It would have been more useful to:
- expand the sample;
- conduct a prospective study to monitor participants over time;
- examine the causal relationship between diet and nutritional status;
- include a non-vegetarian control group;
- use objective measures of food intake;
- evaluate a wider range of nutritional outcomes (vitamins and minerals).
Overall, the study is well-designed and conducted. However, implementing these improvements would strengthen the validity and generalizability of the results.
Round 2
Reviewer 1 Report
Comments and Suggestions for Authors
The authors has addressed my concerns so I have no further comments.